# Structural basis of the mechanism and inhibition of a human ceramide synthase

Tomas C. Pascoa [1]✉, Ashley C. W. Pike [1], Christofer S. Tautermann [2], Gamma Chi [1], Michael Traub[2], Andrew Quigley [1,4], Rod Chalk[1], Saša Štefanić [3,5], Sven Thamm[2], Alexander Pautsch[2], Elisabeth P. Carpenter [1]✉, Gisela Schnapp [2]✉ & David B. Sauer [1]✉

Ceramides are bioactive sphingolipids crucial for regulating cellular metabolism. Ceramides and dihydroceramides are synthesized by six ceramide synthase (CerS) enzymes, each with specificity for different acyl-CoA substrates. Ceramide with a 16-carbon acyl chain (C16 ceramide) has been implicated in obesity, insulin resistance and liver disease and the C16 ceramide-synthesizing CerS6 is regarded as an attractive drug target for obesity-associated disease. Despite their importance, the molecular mechanism underlying ceramide synthesis by CerS enzymes remains poorly understood. Here we report cryo-electron microscopy structures of human CerS6, capturing covalent intermediate and product-bound states. These structures, along with biochemical characterization, reveal that CerS catalysis proceeds through a ping-pong reaction mechanism involving a covalent acyl–enzyme intermediate. Notably, the product-bound structure was obtained upon reaction with the mycotoxin fumonisin B1, yielding insights into its inhibition of CerS. These results provide a framework for understanding CerS function, selectivity and inhibition and open routes for future drug discovery.

Ceramides are the precursors for the synthesis of complex sphingolipids and are bioactive signaling lipids. In particular, ceramides have been proposed as key metabolic sensors to promote fatty acid use and storage during excessive fatty acid availability[1]. Abnormal ceramide accumulation is associated with metabolic dysfunction and elevated levels of ceramides have been observed in obesity-related metabolic disorders such as diabetes, nonalcoholic fatty liver disease and nonalcoholic steatohepatitis (NASH)[2–4].

Ceramides are composed of a sphingosine (d18:1) long-chain base with an N-linked acyl chain, the length of which is critical to the lipids' biological functions and roles in pathophysiology[5]. For example, C16:0 ceramide is the most common ceramide in adipose tissue and its levels are elevated in this tissue of obese humans[2]. In addition, insulin resistance is correlated with plasma C16:0 and C18:0 ceramides, subcutaneous adipose tissue C16:0 ceramides and hepatic C16:0 and C18:0 ceramides[6–8]. Moreover, total ceramides and C16:0, C22:0 and C24:1 dihydroceramides were found to be elevated in the liver of insulin-resistant patients with NASH[3].

In mammals, de novo synthesis of ceramides is preceded by synthesis of dihydroceramides through the N-acylation of the sphingoid long-chain base sphinganine (dihydrosphingosine; d18:0) by one of six ceramide synthases (CerS1–CerS6)[9]. Alternatively, CerS can directly reacylate recycled sphingosine in the salvage pathway[10]. Of these enzymes, recent observations from human and mouse studies

[1]Centre for Medicines Discovery, Nuffield Department of Medicine, University of Oxford, Oxford, UK. [2]Boehringer Ingelheim Pharma, GmbH & Co. KG, Biberach, Germany. [3]Institute of Parasitology, Vetsuisse and Medical Faculty, University of Zürich, Zürich, Switzerland. [4]Present address: Membrane Protein Laboratory, Research Complex at Harwell, Diamond Light Source, Ltd., Harwell Science and Innovation Campus, Didcot, UK. [5]Present address: Nanobody Service Facility, University of Zürich, AgroVet-Strickhof, Lindau, Switzerland. ✉e-mail: tomas.pascoa@cmd.ox.ac.uk; lizcarpen1@gmail.com; gisela.schnapp@boehringer-ingelheim.com; david.sauer@cmd.ox.ac.uk

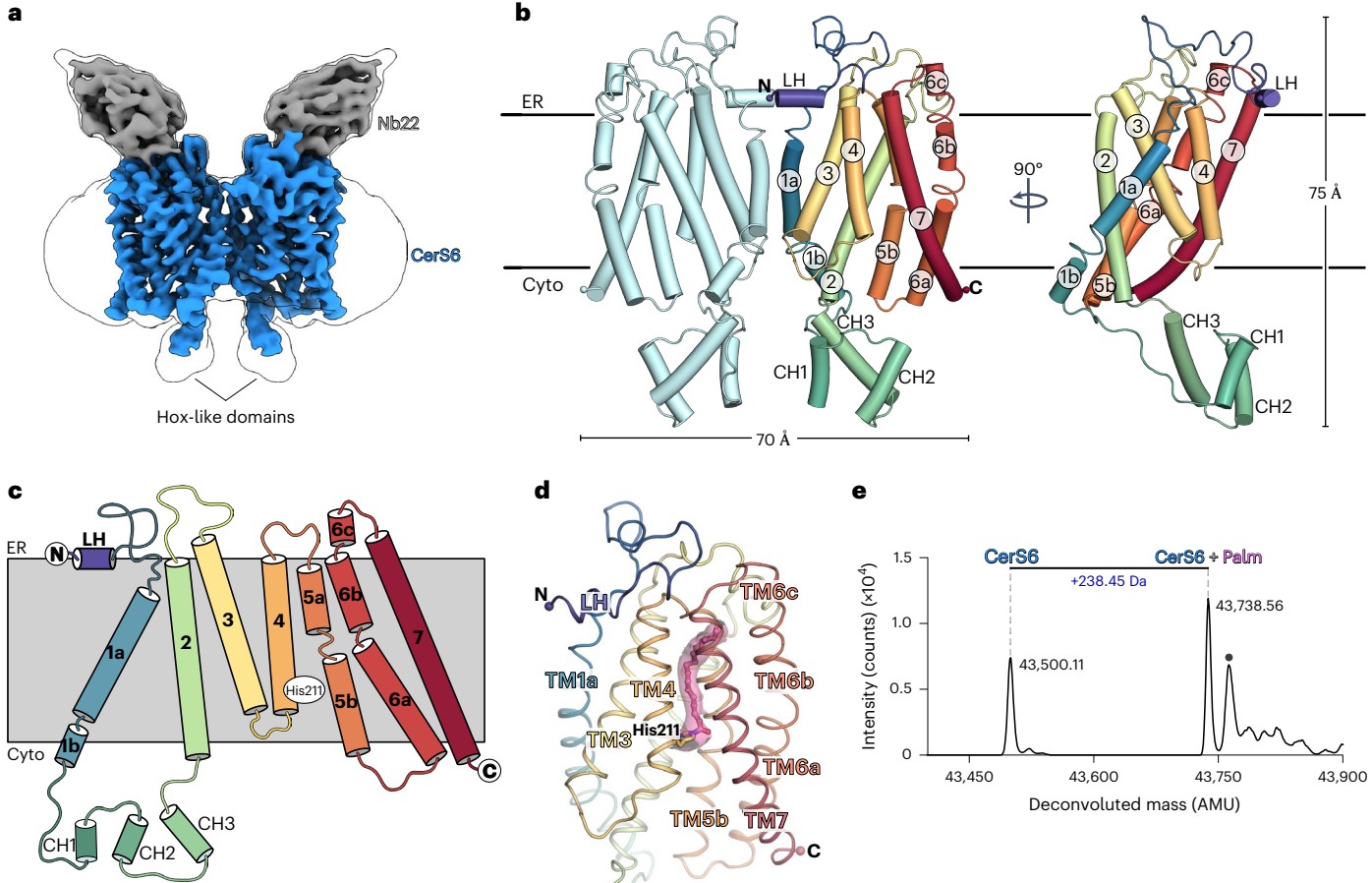

**Fig. 1 | Cryo-EM structure of human CerS6. a**, Cryo-EM map of the CerS6 dimer (blue) with one copy of Nb22 (gray) bound to each CerS6 monomer. **b**, Overall cartoon cylinder representation of the CerS6 dimer structure. One of the monomers is rainbow-colored from purple (N terminus) to red (C terminus). LH, luminal helix. **c**, Schematic representation of the CerS6 seven-TM helix topology. **d**, Cartoon representation of the transmembrane domain of a CerS6 monomer. The covalent acyl–imidazole species is shown in stick representation (acyl chain,

pink carbon atoms) and the Coulombic potential map for this covalent species is shown as a pink transparent surface. The Hox-like domain was omitted for clarity. **e**, Denaturing intact protein MS analysis of purified CerS6 protein, revealing the presence of a covalent modification (+238.45 Da), which matches the expected mass shift corresponding to covalent attachment of a palmitoyl (C16:0) chain. This adduct peak was present in all purifications tested ($n = 6$). AMU, atomic mass units. Cyto, cytoplasm.

highlighted CerS6 as a target for treating obesity-associated metabolic disease, including type 2 diabetes and NASH[2,5,11–13]. Deletion of *CerS6* in mice granted protection against diet-induced obesity, steatohepatitis and insulin resistance[2,13] and liver-specific deletion improved glucose tolerance and mitochondrial morphology[13]. Strikingly, this occurs in CerS6$^{\Delta/\Delta}$ mice but not in CerS5$^{\Delta/\Delta}$ mice, although both show a preference for C16-CoA and, therefore, primarily produce C16 ceramides[14], and hepatic C16:0 ceramide levels were reduced in both knockouts[13]. Resulting from the specific interaction of CerS6-derived C16:0 sphingolipids in mitochondria with the mitochondrial fission factor[13], this revealed that the subcellular localization of ceramide production can lead to drastically different physiological outcomes[13]. In further support of the therapeutic potential of inhibiting CerS6, ablation of the protein's expression in an obese insulin resistance mouse model led to improved body fat, oral glucose tolerance and insulin sensitivity[11].

Central to their enzymatic function, CerSs contain a TRAM–Lag1–CLN8 (TLC) homology domain[15], including a 52-residue Lag1p motif with conserved histidines and aspartates required for activity[16,17]. CerS2–CerS6 also contain a nonessential Hox-like domain between transmembrane (TM) domains 1 and 2 (ref. 18), flanked by two essential and conserved positively charged residues[19]. Regulators of CerS activity include phosphorylation[20], glycosylation[21], dimerization[22] and other protein–protein interactions[23,24]. The six mammalian CerSs also have

different acyl chain-length specificities and tissue expression patterns, further influencing the tissue-specific distribution of the different ceramides[5,9,25]. However, the molecular mechanisms underlying synthesis and regulation of ceramides and dihydroceramides remain unclear.

Despite the interest in pharmacologically reducing CerS6 activity, no specific inhibitors have been described for this family member. The best-characterized CerS inhibitor is fumonisin B$_1$ (FB$_1$), the most prevalent member of the fumonisin family of mycotoxins produced by *Fusarium* species. FB$_1$ is of notable concern as it is a common carcinogenic and teratogenic contaminant of maize, rice, other cereals and cereal-based food stocks[26–28]. FB$_1$ is a potent nonselective inhibitor of all human CerSs, being competitive toward both sphinganine and acyl-CoA substrates[29]. In addition, FTY720 (fingolimod), a prodrug administered to treat multiple sclerosis by modulating sphingosine-1-phosphate receptor 1, is also a nonselective CerS inhibitor[30,31]. Furthermore, a nonphosphorylatable analog of FTY720, P053, was recently shown to potently and selectively inhibit CerS1 (ref. 32), showcasing the possibility of achieving isoform-specific CerS inhibition.

Here, to understand the molecular basis of ceramide and dihydroceramide synthesis by the CerSs, we used a combination of cryo-electron microscopy (cryo-EM), intact protein mass spectrometry (MS) and small-molecule high-resolution MS (HRMS) and tandem MS (MS/MS) to probe the catalytic mechanism of human CerS6.

**Table 1 | Cryo-EM data collection, processing, refinement and validation statistics**

| | CerS6–Nb22 covalent intermediate state (EMD-18770) (PDB 8QZ6) | CerS6–Nb22 N-palmitoyl FB$_1$-bound state (EMD-18771) (PDB 8QZ7) | CerS6–Nb02 covalent intermediate state (EMD-19869) (PDB 9EOT) |
|---|---|---|---|
| **Data collection and processing** | | | |
| Magnification | 130,000 | 130,000 | 130,000 |
| Voltage (kV) | 300 | 300 | 300 |
| Electron exposure (e$^-$ per Å$^2$) | 56.3 | 56.3 | 56.3 |
| Defocus range (μm) | −0.8 to −2.4 | −0.8 to −2.4 | −0.8 to −2.4 |
| Pixel size (Å) | 0.656 | 0.656 | 0.656 |
| Symmetry imposed | C2 | C2 | C2 |
| Initial particle images (no.) | 3,400,118 | 3,998,935 | 4,497,470 |
| Final particle images (no.) | 93,680 | 154,239 | 153,485 |
| Map resolution (Å) | 3.22 | 2.95 | 3.02 |
| FSC threshold | 0.143 | 0.143 | 0.143 |
| Map resolution range (Å) | 2.8–6.5 | 2.6–6.5 | 2.6–6.5 |
| **Refinement** | | | |
| Initial model used | AF2-generated (AF-Q6ZMG9-F1) | AF2-generated (AF-Q6ZMG9-F1) | PDB 8QZ6 |
| Model resolution (Å) | 3.38 | 3.22 | 3.22 |
| FSC threshold | 0.5 | 0.5 | 0.5 |
| Map sharpening B factor (Å$^2$) | −132.5 | −130.7 | −118.3 |
| Model composition | | | |
| Nonhydrogen atoms | 7,226 | 7,366 | 7,214 |
| Protein residues | 906 | 914 | 904 |
| Ligands | 34 | 134 | 34 |
| B factors (Å$^2$) | | | |
| Protein | 79.54 | 82.39 | 66.26 |
| Ligand | 30.50 | 64.67 | 22.90 |
| R.m.s.d. | | | |
| Bond lengths (Å) | 0.002 | 0.003 | 0.002 |
| Bond angles (°) | 0.393 | 0.458 | 0.630 |
| Validation | | | |
| MolProbity score | 1.15 | 1.20 | 1.45 |
| Clashscore | 3.62 | 3.17 | 4.14 |
| Poor rotamers (%) | 0.29 | 0.87 | 0.58 |
| Ramachandran plot | | | |
| Favored (%) | 98.00 | 97.57 | 97.10 |
| Allowed (%) | 2.00 | 2.43 | 2.90 |
| Disallowed (%) | 0 | 0 | 0 |

PDB, Protein Data Bank.

## Results

### Cryo-EM structure of human CerS6

We expressed the human CerS6 protein, truncated at Asp350 to remove the predicted disordered C terminus, and purified the enzyme in glyco-diosgenin (GDN), yielding a mixture of monomeric and dimeric species on size-exclusion chromatography (SEC) (Extended Data Fig. 1a–d). While we selected the dimer species for structural studies to maximize particle size and possible symmetry for single-particle cryo-EM, its relatively low molecular weight (84 kDa) is still challenging for current methods. We, therefore, used a camelid nanobody to solve the structure of a CerS6 dimer at a nominal resolution of 3.2 Å (Fig. 1a,b, Table 1 and Extended Data Fig. 2). The CerS6–Nb22 complex was well resolved throughout the membrane-spanning region but the Hox-like

domain was less well defined. This allowed for unambiguous tracing of CerS6 and Nb22 with the exception of the Hox-like domains, where residues 72–119 were fitted into the envelope of the cryo-EM map as rigid bodies (Methods and Extended Data Figs. 2e and 3a).

The cryo-EM structure reveals a CerS6 homodimer with one nanobody bound per monomer on the luminal face and a lipid adjacent to the dimer interface (Extended Data Fig. 3e and Supplementary Note 1). Each monomer has seven TM helices (Fig. 1b,c), with the dimer interface formed by TM1a, the C-terminal end of TM3 and the TM3–TM4 loop. The TLC domain (TM2–TM7) forms a barrel containing two sets of three-TM units arranged as inverted repeats (Extended Data Fig. 4a), while TM1 is attached to the outside of the barrel through contacts with TM2 and TM3. The TM2–TM7 barrel is assembled around a central cavity, which

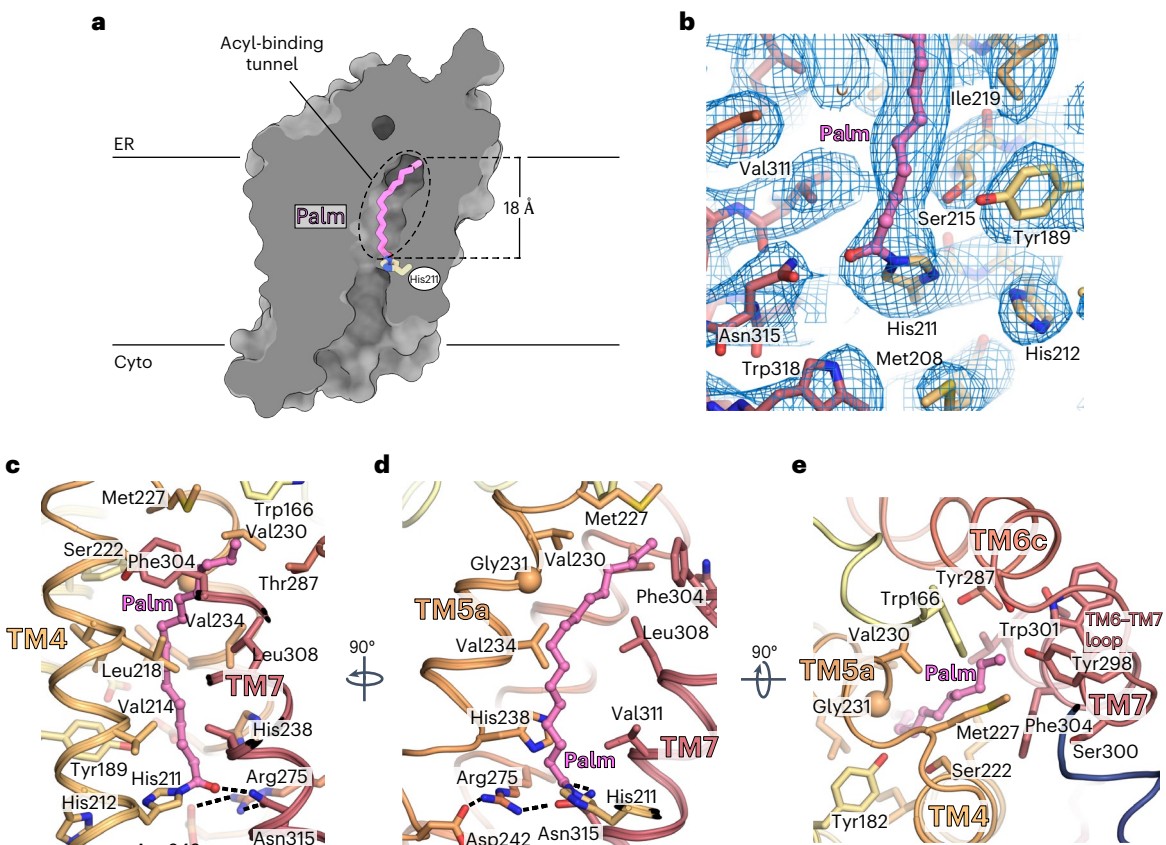

**Fig. 2 | CerS6 contains an acyl chain-binding tunnel buried deep in the membrane. a**, Cutaway molecular surface representation of the CerS6 TM region, revealing the presence of a long central cavity spanning the entire length of the protein. The palmitoyl chain covalently attached to His211 is bound in a narrow tunnel on the ER luminal half of the central cavity. **b**, Coulombic potential map in the region around the covalent linkage of the acyl chain to His211. **c–e**, CerS6's acyl-binding tunnel, viewed from the membrane plane (**c** and **d**) or the ER face (**e**). Side chains lining and capping the tunnel are shown as sticks. Hydrogen bonds in the active site are shown as dashed lines.

is open to the cytoplasm and closed to the endoplasmic reticulum (ER) lumen by a loop between TM6 and TM7. This central cavity has a large cytoplasmic opening (approximately 23 Å × 12 Å), creating a vestibule that funnels toward the midpoint of the membrane, where it forms a ~5-Å-wide tunnel that stretches toward the ER side.

Interestingly, the fold of the TM2–TM7 barrel of CerS6 resembles that of the six-TM barrels found in very-long-chain fatty acid elongase 7 (ELOVL7)[33] and transmembrane proteins (TMEMs) 120A and 120B (refs. 34,35). Additionally, a structurally homologous six-TM barrel is predicted in the AlphaFold2 (AF2) models of 3-hydroxyacyl-CoA dehydratases (HACDs) 1–4 (Extended Data Fig. 4b), which participate in the same four-enzyme acyl-CoA elongation cycle as ELOVLs[36]. The observation that different acyl-CoA-modifying enzymes share a similar six-TM barrel structure is intriguing, suggesting that this architecture is likely optimized for the recognition and reaction with acyl-CoA substrates.

### CerS6 copurifies with a covalently bound C16:0 acyl chain
The cryo-EM density map revealed the presence of a long density extending from near the membrane midpoint to the occluded ER end of the central cavity, spanning the entire length of the narrow tunnel (Figs. 1d and 2a). This density was continuous between protein and the unknown molecule, suggesting a covalent modification by a lipid. To probe the presence of covalent adducts, we performed denaturing intact protein MS analysis of purified protein samples and identified a +238.45-Da mass shift (Fig. 1e and Extended Data Fig. 1f,g). Notably, this adduct mass agrees with a covalently attached C16:0 acyl chain (theoretical: +238.41 Da). Strikingly, the shape and length of the density in the

cryo-EM map is also consistent with a bound C16 acyl chain. A covalent bond is seen linking the acyl chain to the imidazole $N_\varepsilon$ of His211 (TM4) (Fig. 2b and Extended Data Fig. 3b), an absolutely conserved histidine that is required for CerS activity[16,17]. To further confirm this covalent adduct, we determined an additional CerS6 structure in complex with a second nanobody, Nb02, to a nominal resolution of 3.0 Å (Table 1 and Extended Data Fig. 5). The overall CerS6 structure is unchanged between nanobody complexes (TM region backbone root-mean-square deviation (r.m.s.d.) = 0.39 Å), with the improved local resolution in the active site validating the His211–acyl link (Extended Data Fig. 5g). Thus, on the basis of the extended density in the cryo-EM map and the observed mass adduct, we modeled a palmitoyl (C16:0) chain covalently attached to His211.

The acyl chain is surrounded in the narrow tunnel primarily by hydrophobic side chains from TM4, TM5 and TM7 (Fig. 2c–e). The tunnel is sealed at the ER end by the TM6–TM7 loop (Fig. 2e), revealing how the chain-length preference of CerS6 for C16-CoA is determined by limiting the number of carbons that can fit in the acyl-binding tunnel, in agreement with earlier reports that this loop determines CerS acyl chain specificity[21].

### Catalysis by CerS6 proceeds through a ping-pong mechanism
Acyltransferases catalyze two-substrate, two-product reactions that can proceed through either ternary-complex or double-displacement (ping-pong) mechanisms[37]. In a ternary-complex mechanism, both the acyl donor and the acyl acceptor bind simultaneously and the enzyme catalyzes the direct acyl transfer from one substrate to the other. In

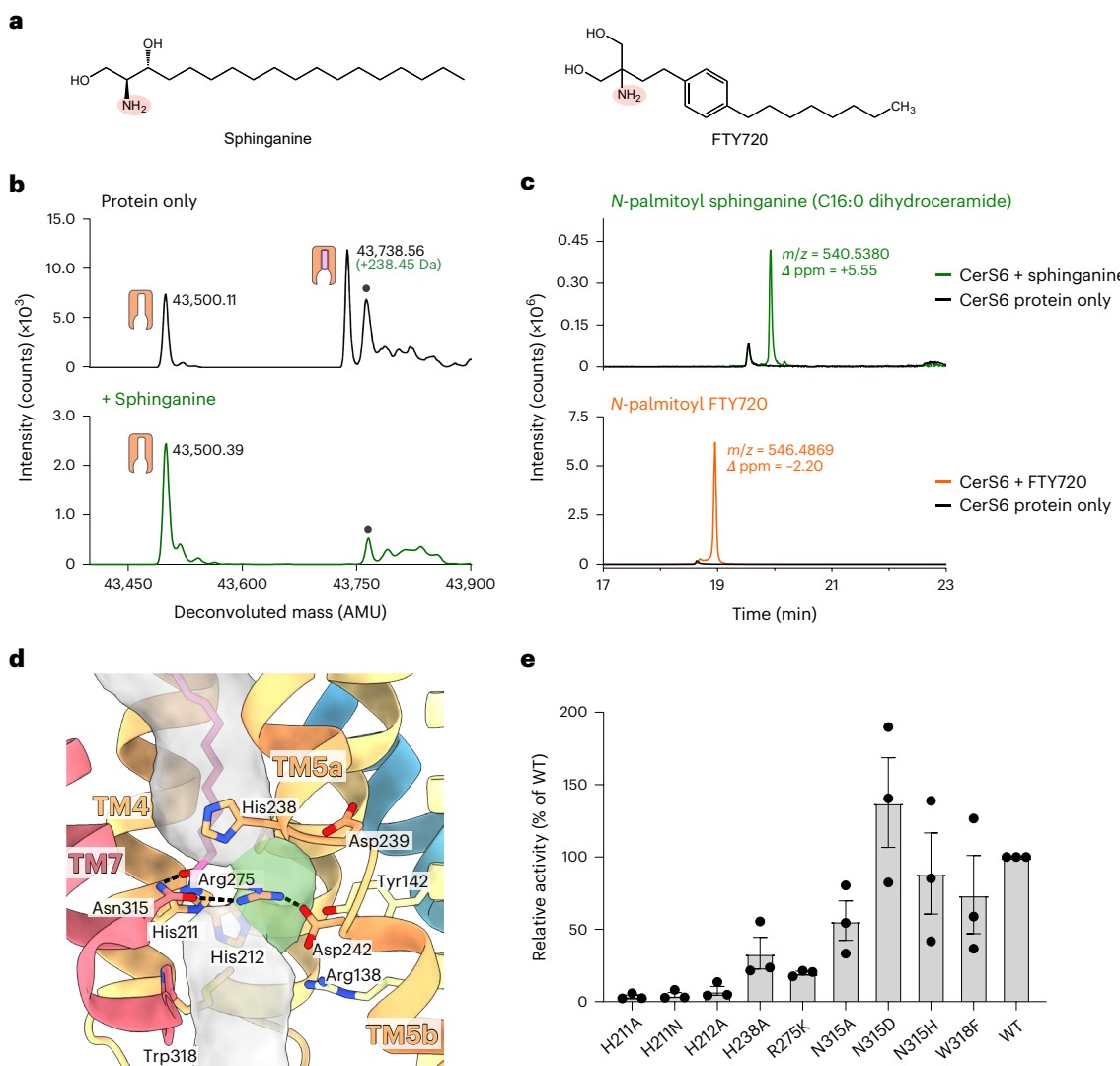

**Fig. 3 | Unraveling the catalytic mechanism of CerS6 using MS. a**, Chemical structures of the natural long-chain base substrate sphinganine (dihydrosphingosine) and the drug FTY720 (fingolimod). The homologous primary amines are highlighted in salmon. **b**, Intact mass analysis of protein samples after incubation in the absence of substrates (black) or in the presence of sphinganine (green) (n = 3 biological replicates). Replicate traces are provided in Extended Data Fig. 6. Deconvoluted mass peaks are indicated as follows: unmodified enzyme, orange icon; covalent acyl–enzyme species, orange and pink icon; background species present in all traces, gray circle. **c**, LC–HRMS detection of the reaction products. EICs are shown for the expected [M + H]⁺ ions of each of the reaction products after incubation of the acyl–enzyme

intermediate with sphinganine (C16:0 dihydroceramide, green) or FTY720 (N-palmitoyl FTY720, orange). EICs obtained after incubation in the absence of substrates are shown in black. **d**, CerS6's active site, viewed from the plane of the membrane. The central cavity is shown as a transparent gray surface, highlighting the presence of a side pocket (highlighted in green) adjacent to the acyl carbonyl. Hydrogen bonds are shown as dashed lines. **e**, Mutational analysis of the CerS6 active site by comparison of the C16:0 dihydroceramide synthase activity of WT and active site mutants (n = 3 independent biological replicates). Gray bars correspond to the mean of the biological replicates. Data points represent each biological replicate, corresponding to the mean of four technical replicates. Error bars show the s.e.m.

contrast, a ping-pong mechanism involves two independent steps. Initially, reaction with the first substrate (the acyl donor) results in the transfer of the acyl chain to a nucleophilic residue, forming a covalent acyl–enzyme intermediate. This acyl chain is then transferred to the second substrate (the acyl acceptor) in the second step of the reaction.

The CerS6 structure has insufficient space for both substrates to access the active site at the same time because of the pathway from the cytoplasm narrowing to ~5 Å before the active site, thereby indicating that a ternary-complex mechanism is unlikely. Furthermore, the C16:0 chain covalently attached to the conserved His211 fits with CerS6's selectivity for palmitoyl-CoA and suggests that this species could correspond to the acyl–enzyme intermediate of a ping-pong type reaction mechanism. To investigate whether the acylated enzyme species corresponds to a real catalytic intermediate, we incubated the purified

protein with the enzyme's second substrate, the long-chain base sphinganine (Fig. 3a). Using denaturing intact protein MS, we observed the complete loss of the C16:0-modified protein after sphinganine addition (Fig. 3b and Extended Data Fig. 6a). Furthermore, HRMS revealed that incubation of the acylated protein with sphinganine led to the production of C16:0 dihydroceramide (observed m/z: 540.5380; theoretical m/z: 540.5350; mass error: +5.55 ppm) (Fig. 3c). Overall, these results demonstrate that sphinganine reacts with the acyl–enzyme intermediate to deacylate the enzyme and generate the expected reaction product of dihydroceramide, thus supporting a ping-pong reaction mechanism for CerS6. Furthermore, the higher mass accuracy of HRMS also validates our identification of CerS6's acyl modification as C16:0. Taken together with the structural data, this implies that His211 acts as the nucleophile in the first step of the reaction.

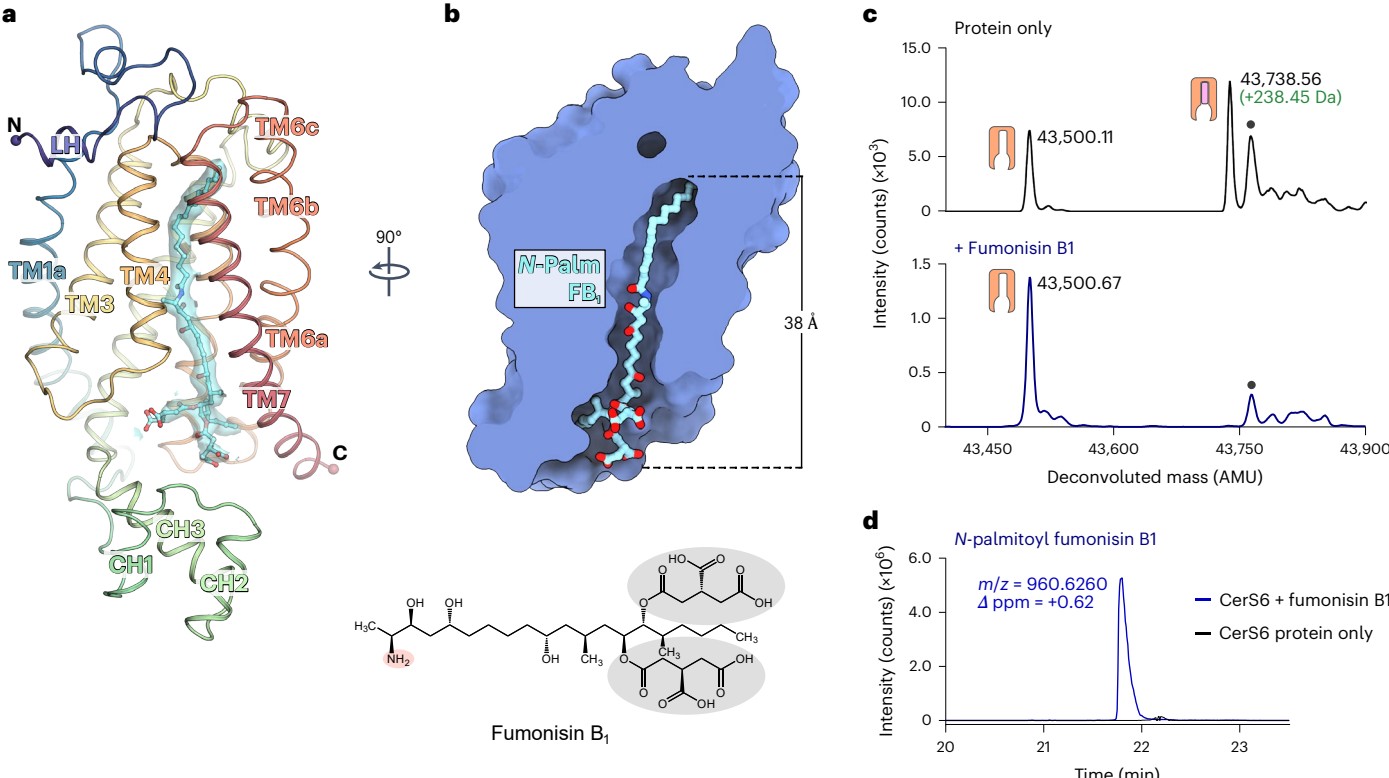

**Fig. 4 | Cryo-EM structure of CerS6 in complex with *N*-palmitoyl FB₁. a**, Cartoon representation of CerS6 with bound *N*-palmitoyl FB₁ (shown as sticks; cyan carbon atoms). The cryo-EM density of the bound product is shown as a transparent cyan surface. **b**, Cutaway molecular surface representation, revealing that the *N*-palmitoyl FB₁ species occupies the entire length of the central cavity. The Hox-like domain was omitted for clarity. **c**, Intact mass analysis of protein samples after incubation in the absence of substrates (black) or in the presence of the mycotoxin FB₁ (blue) ($n = 3$ biological replicates). Replicate traces are provided in Extended Data Fig. 6. **d**, LC–HRMS detection of *N*-palmitoyl FB₁. The EIC for its expected [M + H]⁺ ion is shown after incubation of the acyl–enzyme intermediate with FB₁ (blue) or in the absence of the toxin (black). Inset, chemical structure of FB₁. Its primary amine (salmon circle) and TCA (gray circles) groups are highlighted.

Notably, the CerS substrate sphinganine and FTY720 are chemically similar (Fig. 3a), with potential clinical consequences to drug pharmacokinetics as 15% of FTY720 becomes *N*-acylated in human subjects[38]. Although the major *N*-acyl FTY720 metabolites identified were *N*-stearoyl and *N*-2-hydroxystearoyl FTY720, there is evidence suggesting that a small amount of *N*-palmitoyl (C16:0) FTY720 is formed[38]. Which enzyme catalyzes the *N*-acylation of FTY720 has not been determined but a role for CerS enzymes has been proposed[38]. Incubation of acyl–CerS6 with FTY720 results in a small increase in the protein's thermostability, suggesting that it binds to the purified enzyme (Extended Data Fig. 1e). To investigate whether purified CerS6 can *N*-acylate FTY720 in vitro, we incubated the enzyme with this compound and subsequently identified C16:0 FTY720 in the reaction mixture (observed *m/z*: 546.4869; theoretical *m/z*: 546.4881; mass error: −2.20 ppm) (Fig. 3c), the expected product of a reaction between FTY720 and the CerS6-bound acyl chain. The proposed structure of this C16:0 FTY720 reaction product was further validated on the basis of the product ion MS/MS spectrum of its [M + H]⁺ ion (Extended Data Fig. 6c,d). However, no obvious loss of the acyl–enzyme mass peak was observed when the reaction mixture was analyzed by denaturing intact protein MS (Extended Data Fig. 6b), suggesting that FTY720 is a poor acyl acceptor for CerS6.

### Highly conserved residues line the catalytic site
CerS6's Lag1p motif (Arg202–Tyr253) (TM4–TM5) contains the highly conserved His211/His212 (TM4) and Asp239/Asp242 (L5a-b/TM5), which are suggested to be in the catalytic site and have been shown to be essential for function in mammalian CerS[16] and *Saccharomyces cerevisiae* Lag1p[17].

Within the CerS6 structure, these residues line the central cavity near the midpoint of the membrane (Fig. 3d). Notably, the copurified covalently linked C16:0 chain is covalently attached to the first histidine of the Lag1p motif (His211) (Fig. 2a–d), which appears to act as the nucleophile in the first step of the reaction and is required for CerS6 function (Fig. 3e). Although it is remarkable that CerS6 uses a nucleophilic histidine, as cysteines or serines are more commonly used as nucleophiles, there are multiple well-documented examples of other enzymes with nucleophilic histidines in their catalytic mechanisms[33,39–43].

Histidine nucleophiles are typically oriented and activated by hydrogen bonding between a proton at N$_\delta$ of the imidazole ring and a hydrogen-bond acceptor, ensuring that N$_\varepsilon$ remains unprotonated[33,40,44]. Examining our structure, we noted the proximity of His212 to the acylated His211 in CerS6, suggesting that this residue is the hydrogen-bond acceptor. While the N$_\delta$ of His212 is located 3.84 Å away from the N$_\delta$ of His211, it is plausible that the two side chains interact in the acyl-CoA-bound state. Indeed, we found that substitution of His212 to alanine results in the complete loss of activity (Fig. 3e). Substitution of residues corresponding to His212 of CerS6 also caused loss of function in the homologous mouse and human CerS1 (refs. 16,45) and yeast Lag1p[17]. This demonstrates that the second histidine has a key role in catalysis and explains the pathogenic effect of substituting the equivalent histidine of CerS1 to a glutamine in persons with progressive myoclonic epilepsy and dementia[45]. Moreover, we recently demonstrated that the reaction of human ELOVL7 with acyl-CoA substrates also involves a double-histidine motif, where the first histidine acts as a nucleophile and is activated by hydrogen bonding to the second[33]. Strikingly, structural alignment of CerS6 and ELOVL7 places the histidine

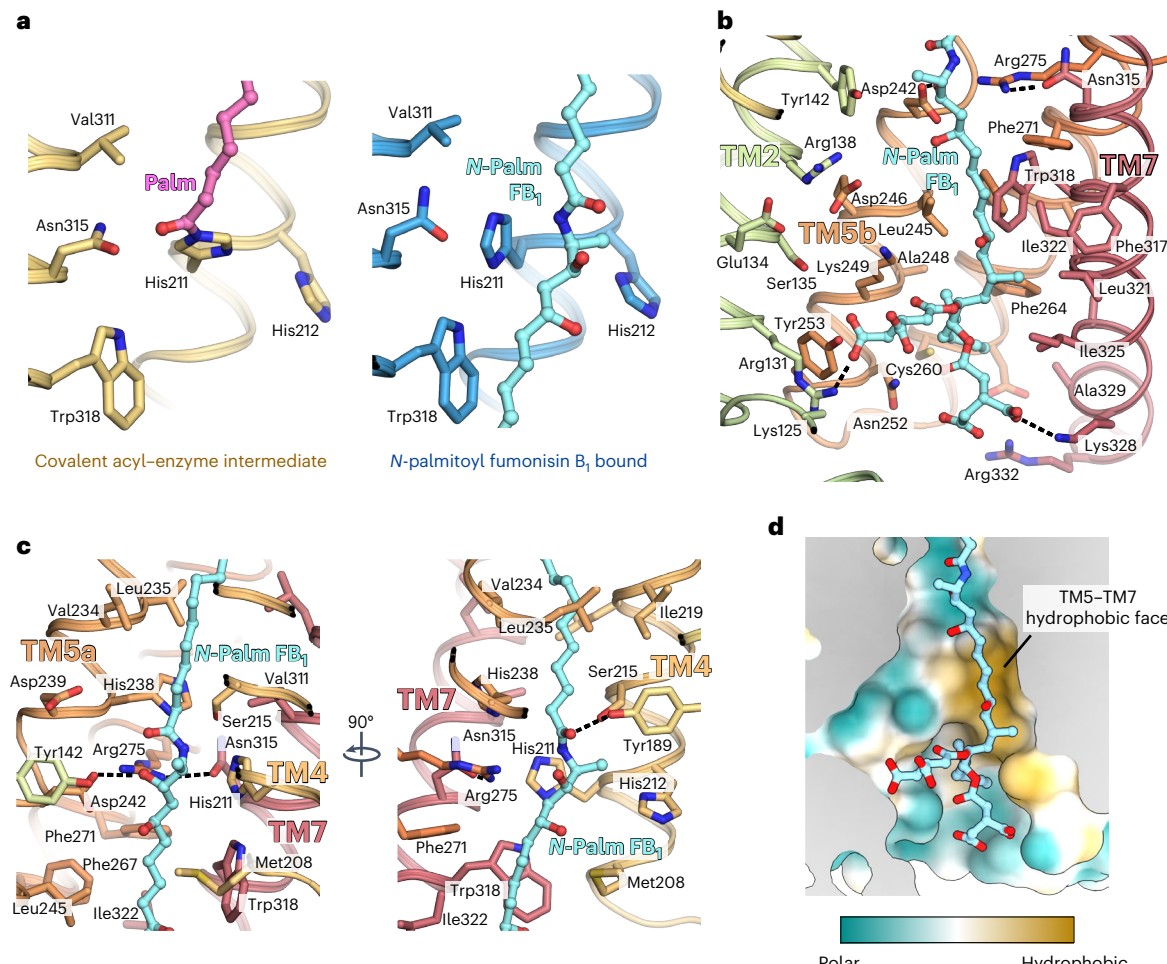

**Fig. 5 | Binding mode of _N_-palmitoyl FB₁. a,** Close-up view of the CerS6 active site in the covalent acyl–enzyme intermediate and _N_-palmitoyl FB₁-bound states, showing the transfer of the palmitoyl chain from His211 to the toxin. **b,** Cytoplasmic portion of the central cavity, viewed from the membrane plane. Residues lining the cavity are shown as sticks. Polar interactions between the carboxylates of the TCA groups of FB₁ and positively charged residues on TM2 and TM7 are shown as dashed lines. **c,** Active site, viewed from the plane of the membrane. **d,** Polar and nonpolar surfaces on the cytoplasmic half of the central cavity. Cutaway molecular surface view, showing that the hydrocarbon chain of FB₁ interacts with the large hydrophobic face formed by TM5–TM7.

pairs in equivalent positions (Extended Data Fig. 4c), suggesting that these residues have similar roles in the first step of their respective reactions.

Formation of the acyl–enzyme intermediate proceeds through an oxyanion transition state, which is usually stabilized by hydrogen bonding. Within the acyl–CerS6 structure, the carbonyl oxygen of the covalently attached C16:0 chain is within hydrogen-bonding distance (2.8 Å) of the N$_\delta$ of Asn315 (TM7). On the basis of the proximity of these moieties, we hypothesized that this interaction stabilizes the formation of the oxyanion transition state during catalysis. Consistent with this notion, an asparigine is present at the equivalent position in CerS1, whereas CerS2–CerS5, Lag1p and Lac1p have a histidine at this position (Extended Data Fig. 7), which could likewise act as a hydrogen-bond donor. Indeed, we found that an Asn315His mutant preserves activity. Surprisingly, however, substitution of Asn315 to an alanine or aspartic acid did not result in loss of activity (Fig. 3e), which does not support our initial hypothesis for the mechanism of oxyanion transition-state stabilization.

On the opposite side of the active site, the essential conserved Asp242 (TM5b) participates in a hydrogen-bonding network with Arg275 (L6a-b) and Asn315 (TM7) (Fig. 3d). We found that an Arg275Lys mutant retained only approximately 20% of wild-type (WT) activity (Fig. 3e), suggesting that the guanidino group of Arg275 is important for function and may facilitate the alignment of neighboring residues in the active site.

There is a third histidine on TM5a that is conserved in mammalian CerS but is a methionine in yeast Lag1p and Lac1p, thus being of ambiguous importance to the enzymes' reaction cycle. In the CerS6 structure, His238 is located above the acyl–imidazole carbonyl and lays against the hydrocarbon chain (Fig. 3d). We found that substitution of this His238 to alanine reduced activity by over 50% (Fig. 3e), thus suggesting that this residue may participate in aligning the acyl chain.

Lastly, we noticed the presence of a side pocket in the active site of CerS6, adjacent to His211 (Fig. 3d) and lined by Tyr142 (TM2), Tyr189 (TM3), His212 (TM4), Asp239 (L5a-b), Asp242 (TM5b) and Arg275 (L6a-b). The position and chemistry of this pocket relative to the carbonyl of the acyl–enzyme intermediate suggest that this is likely where the amino alcohol moiety of the second (long-chain base) substrate binds. Supporting this notion, the residues of this pocket are highly conserved (Extended Data Fig. 7). This pocket, therefore, likely ensures that the primary amine of the long-chain base is ideally located for a nucleophilic attack on the carbonyl carbon of the acyl–imidazole intermediate in the second step of the reaction.

## Structure of _N_-acyl FB₁-bound CerS6

Having captured the intermediate state of the CerS6 enzyme, we next set out to examine the structural basis of CerS inhibition by FB₁. To this end, we incubated purified Nb22-bound acyl–CerS6 with FB₁ and determined the complex's structure to a nominal resolution of 2.95 Å

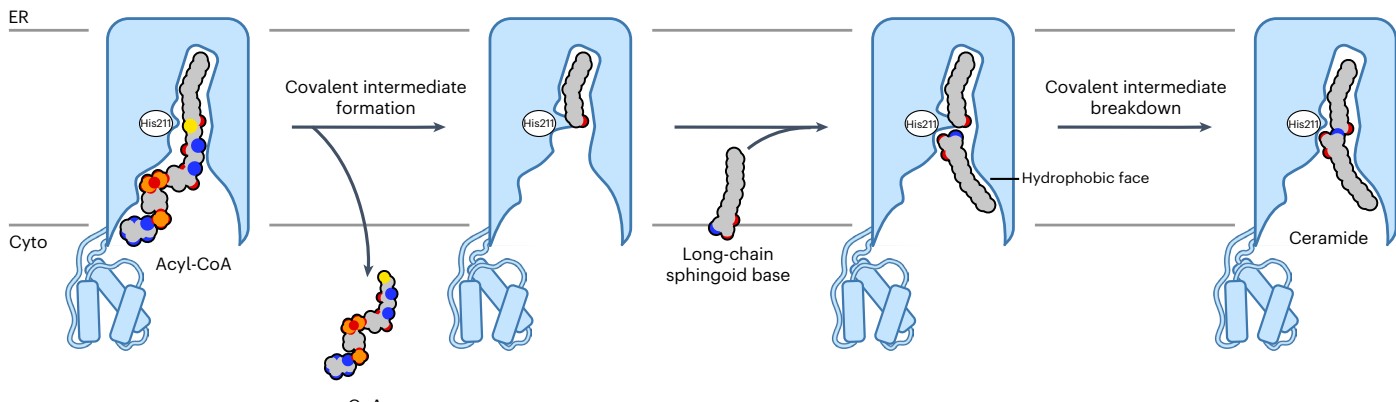

**Fig. 6 | Proposed double-displacement (ping-pong) mechanism of CerSs.** Initially, the acyl-CoA substrate binds with the acyl chain buried within the central tunnel and the CoA moiety sitting near the cytoplasmic entrance to the central cavity. In the first step, the nucleophilic attack of His211 on the acyl-CoA thioester carbonyl results in thioester cleavage, covalent acyl–imidazole intermediate formation and release of CoA. Subsequently, the long-chain

sphingoid base substrate binds with its hydrocarbon chain interacting with the hydrophobic face of the central cavity and its amino alcohol moiety sitting in the side pocket in the active site. In the second step of the reaction, the primary amine of the long-chain base attacks the acyl–imidazole intermediate, leading to covalent intermediate breakdown and formation of the final *N*-acyl sphingoid base (ceramide) product.

(Fig. 4a,b and Extended Data Fig. 8). In comparison to the covalent intermediate state, the overall structure of each CerS6 monomer did not change (TM region backbone r.m.s.d. = 0.39 Å), although the flexible Hox-like domains rotated by approximately 10°.

The cryo-EM map of the putative FB$_1$-bound complex contains a long, continuous nonprotein density starting within the ER luminal end of the narrow tunnel and extending to the cytoplasmic entrance to the central cavity (Fig. 4a and Extended Data Fig. 3c,d). The observed density is too long to be FB$_1$ alone. Notably, incubation of palmitoyl–CerS6 with FB$_1$ completely removed the acyl group from the enzyme (Fig. 4c and Extended Data Fig. 6a) and generated C16:0–FB$_1$ (observed *m/z*: 960.6260; theoretical *m/z*: 960.6254; mass error: +0.62 ppm) (Fig. 4d). Noting that the covalent bond between the enzyme and the palmitoyl group was replaced by a bond to the toxin, we modeled this density as C16:0–FB$_1$ (Fig. 5a and Extended Data Fig. 3c). These results indicate that FB$_1$ reacted with the covalent acyl–enzyme intermediate, whereby the palmitoyl group was released from His211 and attached to the toxin, generating the *N*-acyl FB$_1$ product. This product remains bound to the enzyme, thus acting as an inhibitor. This finding is consistent with earlier reports that CerS enzymes can *N*-acylate FB$_1$ (refs. 46,47).

The *N*-acyl FB$_1$ species spans the entire length of the central cavity, with the acyl chain remaining bound in the narrow tunnel at the occluded ER end of the cavity and the FB$_1$ portion sitting in the wider cytoplasmic entrance (Fig. 4b and Fig. 5b,c). The newly formed amide bond sits in the active site (Fig. 5c), with the carbonyl oxygen forming a hydrogen bond with Tyr189 (TM3) and the amide NH forming a hydrogen bond with the N$_\varepsilon$ of His211 (TM4). The 3-hydroxyl group of FB$_1$, which is also present in sphingoid base substrates, forms a hydrogen bond with the strictly conserved and catalytically essential Asp242 (TM5b)[16,17]. We hypothesize that this interaction with the 3-hydroxyl group of sphingoid base substrates would optimally place the sphingoid primary amine to attack the carbonyl carbon.

Below the active site, the cytoplasmic vestibule of the central cavity has both polar and nonpolar surfaces, where we noted that the hydrocarbon chain of FB$_1$ interacts with the long hydrophobic face formed by TM5–TM7 (Fig. 5b,d). This hydrocarbon chain lies against the conserved Trp318 of TM7 just beneath the active site. Substituting the corresponding residue in CerS5 to an alanine abolishes enzymatic activity[48]. In contrast, a W318F substitution in CerS6 retained approximately WT levels of activity (Fig. 3e). Therefore, we hypothesize that the conserved tryptophan orients the sphingoid base substrate to

place its amino alcohol moiety in the active site's side pocket for the second step of the reaction.

The face of the central cavity opposite the FB$_1$ hydrocarbon-binding surface displays a polar character and is lined by conserved charged residues that may participate in CoA binding (Fig. 5b,d). Accordingly, the positive charges provided by Lys125 and Arg131, located at the cytoplasmic entrance to the cavity, adjacent to the Hox-like domain, are important for the function of CerS5 and CerS6 (ref. 19). Furthermore, substitution of Lys131 in CerS3, equivalent to CerS6's Arg131, to alanine was implicated in autosomal recessive congenital ichthyosis[49], a disorder that can be caused by loss of ≥C26 ceramides because of mutations in *CERS3* (refs. 50,51).

The two tricarballylic acid (TCA) groups of FB$_1$ span the width of the cytoplasmic end of the central cavity, where one of the groups interacts with Arg131 (TM2) and the other interacts with Lys328 (TM7) (Fig. 5b). Thus, through these charge–charge interactions, FB$_1$ links the first helix of the first three-TM bundle and the last helix of the second. Supporting the notion that FB$_1$ acts to restrain the structure, atomistic molecular dynamics (MD) simulations of covalent intermediate and *N*-acyl FB$_1$-bound CerS6 monomers indicated that a bound *N*-palmitoyl FB$_1$ reduces overall protein flexibility, including TM6, TM7 and the Hox-like domain (Extended Data Fig. 9a–c). Moreover, while Lys125 was not resolved in our cryo-EM maps, in the simulations, Lys125 formed a salt bridge with the same TCA group that contacts Arg131 (Extended Data Fig. 9d). Furthermore, the simulations showed that this TCA also contacts Lys249 (TM5b), Tyr253 (TM5b) and Arg118 and Arg121 (last helix of the Hox-like domain). Therefore, FB$_1$ appears to mimic several CoA interactions through the binding of one of its TCA moieties at the polar face of the cytoplasmic cavity. Lastly, on the opposite side of the cavity, the other TCA moiety can form salt bridges with Lys328 or Arg332 (TM7) (Extended Data Fig. 9d). Overall, our cryo-EM structure and simulations revealed that FB$_1$ is anchored by polar interactions at opposite sides of the six-TM barrel, which likely hinder product release, arresting the enzyme in an inhibitory product-bound state.

## Discussion

CerSs carry out an essential step in sphingolipid biosynthesis and are promising drug targets. Yet, our limited understanding of their structures and biochemical mechanism has hindered their therapeutic targeting. Here, we provide structural snapshots of CerS6 at two stages of its reaction cycle, revealing its ping-pong (double-displacement)

reaction mechanism that uses a histidine nucleophile to attack the acyl-CoA thioester to form a stable acyl–imidazole intermediate. This intermediate subsequently reacts with the sphingoid long-chain base substrate to yield the final ceramide or dihydroceramide reaction product (Fig. 6). This sequential reaction and single substrate entry path contrasts with the ternary-complex reaction mechanism and lateral entry gate proposed for CerS2 and Lac1p (Supplementary Discussion).

Although CoA and the hydrocarbon chain of long-chain bases likely interact at distinct faces of the cytoplasmic vestibule, there is not enough space for the acyl-CoA and long-chain base substrates to bind at the same time. Therefore, upon thioester cleavage and covalent intermediate formation, the CoA product needs to exit the central cavity before the long-chain base substrate can bind. We hypothesize that the amino alcohol group of the long-chain base would bind in a discrete side pocket lined by polar residues conserved across the entire CerS family. This long-chain base is further oriented for the second step of the reaction by hydrophobic residues located at the entrance to this pocket, particularly by the conserved Trp318 (TM7). The primary amine of the long-chain base then attacks the acyl–enzyme intermediate to form the final reaction product. Release of this product into the membrane likely occurs between two TM helices and the two halves of the six-TM barrel immediately suggest an egress route between TM4 and TM7. Indeed, we found that the TCA groups of $N$-palmitoyl FB$_1$ attach to both halves of the barrel and restrict the movement of TM7, suggesting that these interactions likely prevent that product's release. This proposed role for the TCA groups of FB$_1$ is also consistent with previous reports that the hydrolyzed form of FB$_1$, lacking the TCA moieties, is a weaker CerS inhibitor[52]. However, FB$_1$ is competitive toward both substrates[29]. This suggests that FB$_1$'s mode of inhibition of CerS is likely to be multifaceted, sterically blocking substrate binding to the apo state and locking the protein in a product-bound state as shown here.

Lastly, the acylation of FTY720 has important consequences for the clinical use of compounds that mimic sphingoid bases. The action of CerS partially explains the $N$-acylated FTY720 species found during the FTY720 drug trials[38] and, by extension, this enzyme family may have a role in the pharmacokinetics of other sphingoid analogs. Overall, our findings support the notion that unintended substrates can lead to the formation of CerS-inhibiting products, as shown by FB$_1$'s ability to arrest the protein in a product-bound state (Extended Data Fig. 10). This suggests that ceramide or $N$-acyl FB$_1$ mimetics could provide novel chemical scaffolds for new CerS inhibitors.

## Online content

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

## Methods

### CerS6 cloning and expression

The *Homo sapiens* CERS6 gene, encoding the CerS6 protein (Uniprot Q6ZMG9), was cloned into the baculovirus transfer vector pHTBV1.1-CT10H-SIII-LIC (adapted from the BacMam vector pHTBV1.1; provided by F. Boyce, Massachusetts General Hospital) upstream of C-terminal tobacco etch virus (TEV)-cleavable 10xHis and Twin-Strep tags. The construct used for structural determination lacked the final 42 amino acids, which are predicted to be disordered. For in vitro biotinylation, the Avi-tagged CerS6 construct contained a Gly-Gly-Gly-Ser linker between CerS6 and the Avi tag, located upstream of the TEV cleavage site. Site-directed mutagenesis was performed using the Q5 site-directed mutagenesis kit (New England Biolabs).

Baculoviral DNA was generated by transposition of DH10Bac with the baculovirus transfer vector. Baculovirus was produced by transfecting *Sf*9 cells (Thermo Fisher Scientific) with the baculoviral DNA using the Insect GeneJuice transfection reagent (Merck Milipore). The virus was amplified by infecting *Sf*9 cells in the presence of 2% FBS and incubating for 72 h at 27 °C. For large-scale protein expression in Expi293F GnTI⁻ cells (Thermo Fisher Scientific) in Freestyle 293 expression medium (Thermo Fisher Scientific), cells were transduced with baculovirus and supplemented with 5 mM sodium butyrate. Cells were incubated in a humidity-controlled orbital shaker at 37 °C with 8% $CO_2$ for 48 h and subsequently harvested by centrifugation at 900$g$ for 15 min. Pelleted cells were washed with PBS, pelleted again by centrifugation, flash-frozen in liquid $N_2$ and stored at −80 °C. To compare mutant activity, WT and mutant proteins were expressed in Expi293F cells (Thermo Fisher Scientific) using the same protocol.

### Large-scale CerS6 purification

CerS6-overexpressing cells were resuspended in buffer A (50 mM HEPES pH 7.5, 200 mM NaCl and 5% (v/v) glycerol) and solubilized with 1% (w/v) lauryl maltose neopentyl glycol (LMNG) (Anatrace) and 0.1% (w/v) cholesteryl hemisuccinate (CHS) (Sigma-Aldrich). Insoluble material was removed by centrifugation at 35,000$g$ and the protein was purified from the supernatant by batch binding to Strep-Tactin XT Superflow resin (IBA Lifesciences). The resin was washed initially with buffer A containing 0.05% (w/v) GDN and subsequently with buffer A supplemented with 1 mM adenosine triphosphate (ATP; Sigma-Aldrich), 10 mM $MgCl_2$ and 0.05% (w/v) GDN. CerS6 was eluted from the resin using buffer A supplemented with 100 mM D-biotin (Fluorochem) and 0.02% (w/v) GDN, while the C-terminal tag was cleaved by TEV protease digestion overnight. The His-tagged TEV protease was removed by binding to $Co^{2+}$-charged TALON resin (Clontech) and the flowthrough was concentrated using a concentrator with a 100-kDa molecular weight cutoff (Corning). The protein was finally purified by SEC on a Superdex 200 Increase 10/300 GL column (GE Healthcare), pre-equilibrated in SEC buffer (20 mM HEPES pH 7.5, 200 mM NaCl and 0.01% (w/v) GDN).

For nanobody generation and selection, GDN was replaced in the purification buffers with 0.003% (w/v) LMNG and 0.0003% (w/v) CHS. For in vitro biotinylation of C-terminally Avi-tagged CerS6, during the TEV cleavage step, the protein sample was further supplemented with 15 mM $MgCl_2$, 15 mM ATP, 50 mM bicine pH 8.3 and BirA at a 1:15 (w/w) BirA:CerS ratio. Biotinylation efficiency was routinely monitored by denaturing intact protein MS.

For structural determination, CerS6 was mixed with a 1.5× molar excess of nanobody (either Nb22 or Nb02) and incubated for 1 h on ice. The CerS6–nanobody complexes were purified by SEC, concentrated to 5 mg ml⁻¹ using a concentrator with a 100-kDa molecular weight cutoff (Sartorius) and processed immediately for cryo-EM.

### Nanobody library generation and selection

Purified CerS6 was reconstituted at a 1:20 (protein:lipid (w/w)) ratio into liposomes composed of 1-palmitoyl-2-oleoyl-*sn*-glycero-3 -phosphocholine (POPC), 1-palmitoyl-2-oleoyl-*sn*-glycero-3 -phosphoglycerol, 1,2-dimyristoyl-*sn*-glycero-3-phosphocholine and 1,2-dimyristoyl-*sn*-glycero-3-phosphoglycerol (7:3:7:3 (w/w)) as previously described[53].

To obtain anti-CerS6 nanobodies, alpacas were immunized and the nanobody library was generated as previously described[54], with the exception that 200 μg of purified CerS6 in proteoliposomes was used for each immunization. All the procedures concerning alpaca immunization were approved by the Cantonal Veterinary Office of Zurich (license no. ZH 198/17). The resulting nanobody library was screened by biopanning against CerS6. Subsequently, 190 single clones from the enriched nanobody library were analyzed by ELISA for binding to CerS6. A total of 96 ELISA-positive clones were Sanger-sequenced and grouped in families according to their complementarity-determining region length and sequence diversity[55]. Of these, 42 unique nanobodies were identified as belonging to 26 nanobody families and were taken for further validation.

Nanobodies were expressed in 50-ml scale in WK6 cells and purified from periplasmic extracts using $Ni^{2+}$-NTA resin, as previously described[55]. Unique nanobodies were screened by biolayer interferometry (BLI) in an Octet Red 384 system (Sartorius) using Streptavidin SA biosensors (Sartorius) loaded with 40 μg ml⁻¹ biotinylated CerS6 in SEC buffer containing 0.003% (w/v) LMNG and 0.0003% (w/v) CHS. Nanobodies with slow off rates were identified and nanobodies Nb22 and Nb02 were prioritized for structural studies.

### Thermal stability measurements

Thermal unfolding experiments were carried out by measuring intrinsic tryptophan fluorescence on a Prometheus NT.48 instrument (NanoTemper Technologies) using 5 μM CerS6 solubilized in 0.003% LMNG and 0.0003% CHS. The protein was incubated for 1 h at 4 °C in the presence or absence of 20 or 100 μM FTY720 (Sigma-Aldrich) or $FB_1$ (Sigma-Aldrich). Proteins were heated from 20 °C to 95 °C, at a rate of 1 °C min⁻¹, and unfolding was monitored by the ratio of fluorescence emission at 350 nm and 330 nm. The melting temperature was determined from the inflection point of the transition using the PR.ThermControl software (NanoTemper Technologies).

### Cryo-EM sample preparation and data acquisition

The purified CerS6–Nb22 and CerS6–Nb02 complexes (5 mg ml⁻¹) were applied to freshly glow-discharged Quantifoil 200-mesh Au R1.2/1.3 grids. Plunge-freezing in liquid ethane was carried out using a Vitrobot Mark IV (Thermo Fisher Scientific) set to 4 °C and 100% humidity. For the CerS6–Nb22–$FB_1$ complex, the SEC-purified CerS6–Nb22 complex was incubated with 120 μM $FB_1$ (Sigma-Aldrich) for 10 min on ice before protein concentration and subsequently for an additional 2 h on ice before grid preparation. EM grids were screened on a Glacios (Oxford Particle Imaging Center (OPIC)) and high-resolution data were collected on a Titan Krios G3 microscope (Leicester Institute of Structural and Chemical Biology (LISCB)) operating at 300 kV and equipped with a Bioquantum energy filter (Gatan) (operated at 20-eV slit width) and a K3 direct electron detector (Gatan) at ×130,000 nominal magnification, in super-resolution mode (2× binning; physical pixel size: 0.656 Å per pixel), using a defocus range between −0.8 μm and −2.4 μm. The total exposure dose was 56.3 e⁻ per Å², fractionated over 50 frames. In total, 14,309, 18,386 and 14,656 videos were collected for the CerS6–Nb22, CerS6–Nb22–$FB_1$ and CerS6–Nb02 datasets, respectively.

### Cryo-EM data processing and model building

Movies were motion-corrected using MotionCor2 (ref. 56) and contrast transfer function (CTF) estimation was carried out in cryoSPARC[57] (version 3.3.1). Micrographs with bad CTF fitting (>5 Å) were excluded, yielding 14,196, 16,647 and 14,596 micrographs for further analysis for the CerS6–Nb22, CerS6–Nb22–$FB_1$ and CerS6–Nb02 datasets, respectively.

For the CerS6–Nb22 covalent acyl–enzyme intermediate dataset, particles were initially blob-picked and extracted from a subset of micrographs; then, representative well-resolved two-dimensional (2D) classes were used for template-based picking of the entire dataset. A total of 3,400,118 template-picked particles were extracted in a 360-pixel box Fourier-cropped to 120 pixels, corresponding to a pixel size of 1.968 Å per pixel, and classified in two rounds of reference-free 2D classification yielding 1,261,928 good particles. Four ab initio models were generated and used as reference models in heterogeneous refinement. Particles belonging to the well-resolved 2:2 CerS6–Nb22 dimer complex class (53% of input particles) were selected and subjected to a further round of heterogeneous refinement. The best-resolved particles (497,504) were then subjected to nonuniform refinement[58] with C2 symmetry imposed, yielding a 3.21-Å reconstruction (Fourier shell correlation (FSC) = 0.143). These aligned particles were imported into RELION using the csparc2star.py script from the University of California, San Francisco pyem package[59] and re-extracted in a 360-pixel box Fourier-cropped to 270 pixels, corresponding to a pixel size of 0.875 Å per pixel. In RELION, CTF parameters were refined, followed by Bayesian particle polishing[60]. The polished particles were subjected to three-dimensional (3D) classification without image alignment (k = 10, T = 12). Certain classes displayed worse side-chain density and local deviations from C2 symmetry. Therefore, to improve map quality, the 93,680 particles belonging to the two highest-resolution C2-symmetric classes were reimported into cryoSPARC and subjected to a final nonuniform refinement with C2 symmetry, yielding a 3.22-Å reconstruction (FSC = 0.143). Importantly, even though CTF refinement, Bayesian polishing and 3D classification in RELION did not improve the nominal resolution, the final reconstruction revealed better-defined features, including improved side-chain density in the CerS6 active site (Extended Data Fig. 2f,g).

The CerS6–Nb22 N-acyl FB₁ dataset was processed using the same workflow as the covalent acyl–enzyme intermediate dataset, with some changes. Briefly, 3,998,935 template-picked particles were classified in two rounds of reference-free 2D classification, yielding 1,353,329 good particles. Four ab initio models were generated and used as reference models in heterogeneous refinement. A total of 641,462 particles belonging to the well-resolved 2:2 CerS6–Nb22 dimer complex class (48% of input particles) were subjected to nonuniform refinement with C2 symmetry imposed, yielding a 3.17-Å reconstruction. After CTF refinement, Bayesian polishing and 3D classification without image alignment in RELION, the 154,239 particles belonging to the three highest-resolution C2-symmetric classes were selected. These particles were reimported into cryoSPARC and subjected to a final nonuniform refinement with C2 symmetry, yielding a 2.95-Å reconstruction (FSC = 0.143).

The CerS6–Nb02 covalent acyl–enzyme intermediate dataset was processed similarly to the other datasets, with some modifications. Briefly, 4,497,470 template-picked particles underwent two rounds of reference-free 2D classification, leading to the identification of 1,025,813 good particles. Following heterogenous refinement, 507,128 particles belonging to the well-resolved 2:2 CerS6–Nb02 dimer complex class (50% of input particles) were subjected to nonuniform refinement with C2 symmetry imposed, yielding a 3.22-Å reconstruction. Particles were imported into RELION and re-extracted in a 432-pixel box Fourier-cropped to 324 pixels, corresponding to a pixel size of 0.875 Å per pixel. After CTF refinement, Bayesian polishing and 3D classification without alignment in RELION, the 153,485 particles belonging to the two highest-resolution C2-symmetric classes were selected. These particles were then reimported into cryoSPARC for a final nonuniform refinement with C2 symmetry applied, yielding a 3.02-Å reconstruction (FSC = 0.143).

Atomic models were generated by fitting the AF2 (ref. 61) prediction of CerS6 (AF-Q6ZMG9-F1) and Phyre2 (ref. 62) nanobody homology models into the cryo-EM maps and subsequently manually adjusted

in Coot[63]. In all datasets, residues 72–119, corresponding to the Hox-like domains, were poorly resolved in the sharpened maps. However, their position was evident in the unsharpened and blurred maps (Extended Data Figs. 2e and 3a). To model this region, we used tight restraints to the AF2 prediction for this domain and docked it into the envelope of the blurred maps ($B_{blur}$ = 200 Å²) and surface-exposed side chains were truncated at Cβ. The atomic models were refined using PHENIX real-space refinement[64] with secondary structure and Ramachandran restraints and noncrystallographic symmetry constraints. Restraint dictionaries for FB₁ and the covalently attached C16:0 chain were generated using AceDRG[65]. The final models comprised residues 2–330 (CerS6–Nb22 and CerS6–Nb02 covalent intermediate state) or 2–334 (CerS6–Nb22 N-palmitoyl FB₁-bound state) of CerS6, residues 1–124 of Nb22 or residues 1–123 of Nb02, the C16:0 chain covalently attached to His211 of CerS6 or the N-palmitoyl FB₁ product, one POPC molecule and the first N-acetylglucosamine (GlcNac) residue of the N-linked glycan visible on Asn18. Weighted $F_o − F_c$ ligand difference maps were calculated using Servalcat[66], available in CCPEM[67], by omitting the ligands from the atomic models. All descriptions and figures are based on the structures of the CerS6–Nb22 complex unless otherwise stated. Structural similarity to other acyl-CoA-binding proteins was identified using DALI[68].

## Denaturing intact protein MS

All MS experiments were carried out on purified CerS6 in the absence of nanobodies. The intact masses of purified protein samples were analyzed by denaturing intact protein MS, conducted using an Agilent 1290 Infinity liquid chromatography (LC) system in line with an Agilent 6530 accurate-mass quadrupole time-of-flight MS instrument (Agilent Technologies), as previously described[33]. Typically, 5–8 μg of purified protein (at 1.5–2.0 mg ml⁻¹), diluted to 20 μl in 30% methanol in 0.1% formic acid, was used per injection. Data were acquired between 100 and 3,200 m/z and analyzed using MassHunter Qualitative Analysis version B.07.00 (Agilent Technologies) software. Peaks between 650 and 3,200 m/z in the sum of the mass spectra obtained during protein elution were deconvoluted using the maximum entropy charge deconvolution algorithm.

The identity of the deconvoluted mass peaks was assigned initially on the basis of the expected mass of the purified proteins and subsequently the observed mass shifts between peaks in the deconvoluted mass spectra. The lower-molecular-weight CerS6 peak corresponded to loss of the initiator methionine (theoretical mass shift of −131.20 Da), acetylation of the new N terminus (theoretical mass shift of +42.04 Da) and addition of an N-linked GlcNAc (theoretical mass shift of +203.19 Da). However, the major glycosylation species observed contained the complete core N-linked glycan (theoretical mass shift of +1217.05 Da), as expected for protein expressed in Expi293F GnTI⁻ cells. In addition, we observed mass shifts of +237.47 and +238.51 Da relative to these two glycosylated species, respectively, which we interpreted as palmitoylation at a single site (theoretical mass shift of +238.41 Da), corresponding to the covalent acyl–imidazole intermediate observed in the cryo-EM structure. One additional modification was observed (approximately +264 Da) but its identity could not be assigned. This unknown modification occurred at a distinct site to that of the palmitoylation, as both modifications could occur simultaneously in the same protein molecule (Extended Data Fig. 1f,g), thus excluding the possibility of this unknown modification occurring in the active site.

To monitor the reaction of the covalent intermediate species with the second substrates, before denaturing intact protein MS analysis, the protein (2 mg ml⁻¹) was incubated with 200 μM sphinganine (Avanti Polar Lipids), 200 μM FB₁ (Sigma-Aldrich) or 600 μM FTY720 (Sigma-Aldrich) for 90 min at 37 °C. All intact mass experiments were conducted at least twice using distinct biological samples. Replicate deconvoluted mass spectra are shown in Extended Data Fig. 6.

## Product detection by LC–HRMS

Product detection by LC–HRMS was conducted on a nanoElute LC system in line with a timsTOF Pro 2 MS instrument (Bruker). The reactions were set up as described for the denaturing intact protein MS experiments, diluted 1:40 (v/v) in 30% methanol in 0.1% formic acid, and 1 µl of each sample was injected onto an IonOpticks C18 nano ultrahigh-performance LC column (1.6-µm particle size; 0.075 mm × 250 mm).

The flow rate was set to 0.5 µl min$^{-1}$ and the solvent system consisted of 0.1% Optima LC–MS-grade formic acid (Fisher Chemical) in high-performance LC electrochemical-grade water (Fisher Chemical) (solvent A) and 0.1% formic acid in Optima LC–MS-grade methanol (Fisher Chemical) (solvent B). The initial condition was 60% solvent B and a linear gradient from 60% to 95% solvent B was applied over 17.8 min to elute the samples. This was then followed by a final 2.2-min isocratic elution with 95% solvent B before the system was re-equilibrated between samples for 5 min with 60% solvent B.

The MS instrument was operated in positive ion mode with a capillary voltage of 1,600 V and the drying gas was supplied at 180 °C with a flow rate of 3 L min$^{-1}$. Additional parameters were as follows: deflection delta, 70 eV; funnel 1 radiofrequency (RF), 350 Vpp; funnel 2 RF, 600 Vpp; multipole RF, 500 Vpp. Data were acquired between 150 and 2,200 $m/z$ and analyzed using the Bruker Compass DataAnalysis 5.3.556 software. The extracted ion chromatograms (EICs) are presented in Figs. 3c and 4d and correspond to the theoretical [M + H]$^+$ ions (tolerance: ±0.005 $m/z$) of the reaction products $N$-palmitoyl dihydrosphingosine (C16:0 ceramide) (theoretical $m/z$: 540.5350), $N$-palmitoyl FB$_1$ (theoretical $m/z$: 960.6254) and $N$-palmitoyl FTY720 (theoretical $m/z$: 546.4881).

## LC–electrospray ionization (ESI)-MS/MS characterization of $N$-palmitoyl FTY720

To structurally characterize the proposed $N$-palmitoyl FTY720 reaction product, purified protein was incubated with 600 µM FTY720 as before and the reaction mixture was initially analyzed by LC–ESI-MS on an Agilent 1290 Infinity LC system in line with an Agilent 6530 accurate-mass quadrupole time-of-flight MS instrument (Agilent Technologies) as described above. This enabled the identification of the putative [M + H]$^+$ ion of the $N$-palmitoyl FTY720 product (observed $m/z$: 546.4849; theoretical $m/z$: 546.4881). Its product ion spectrum was then obtained by LC–ESI-MS/MS. For this purpose, the MS instrument was operated in positive ESI mode (4 GHz). MS parameters were as follows: capillary voltage, 4,000 V; fragmentor voltage, 175 V. Data were acquired between 100 and 1,700 $m/z$. The targeted parent ion (546.4881 $m/z$; retention time, 8.713 min) was fragmented using a collision energy of 14 V.

## Dihydroceramide synthase activity measurements

For activity assays, WT or mutant CerS6 proteins were overexpressed in Expi293F cells and membranes were prepared as follows. The cell pellet from 0.5 L of culture was thawed in PBS and lysed using an Emulsiflex C5 homogenizer (Avestin); then, cell debris was removed by centrifugation. Membranes were subsequently isolated by ultracentrifugation at 160,000$g$ for 90 min, resuspended in assay buffer (20 mM HEPES pH 7.5, 25 mM KCl, 1 mM MgSO$_4$ and 0.1% (v/v) glycerol), flash-frozen in liquid N$_2$ and stored at −80 °C.

On the day of the assay, membranes were thawed and diluted to 0.25 mg ml$^{-1}$. Next, 20 µl of diluted membranes were dispensed per well on flat-bottom polystyrene 384-well Lumitrac plates (Greiner Bio One). Then, 50 µM sphinganine (2.5 µl of 500 µM sphinganine in assay buffer containing 10% ethanol) and 50 µM palmitoyl-CoA (Sigma-Aldrich) (2.5 µl of 500 µM palmitoyl-CoA in assay buffer) were added to each well for a final reaction volume of 25 µl. For untreated controls, 5 µl of assay buffer were added instead of the substrates. The plates were then incubated for 1 h at room temperature and the

reaction was terminated by the addition of 40 µl of butanol spiked with $N$-palmitoyl(d9) dihydrosphingosine (Avanti Polar Lipids) to yield a final concentration of 5 µM $N$-palmitoyl(d9) dihydrosphingosine as an internal standard. The plates were then shaken at 1,800 rpm for 2 min and centrifuged at 1,000$g$ for 30 s. Finally, 40 µl of the organic (upper) phase was transferred into 384-well polypropylene deep-well plates (Greiner Bio One) and diluted with 40 µl of butanol.

The analytical sample handling was performed by a rapid-injecting RapidFire autosampler system (Agilent) coupled to a triple-quadrupole MS instrument (Triple Quad 6500, AB Sciex) as previously described[69], with some modifications. Briefly, the liquid sample was aspirated by a vacuum pump into a 10-µl sample loop for 6,000 ms and subsequently flushed for 3,000 ms onto a C4 cartridge (Agilent) with the aqueous mobile phase (99.5% water, 0.49% acetic acid and 0.01% trifluoroacetic acid; flow rate, 1.5 ml min$^{-1}$). The multiple reaction monitoring transition for C16:0 dihydroceramide is 540.5 → 266.3 $m/z$ (declustering potential, 130 V; collision energy, 38 V) and that for the internal standard C16:0 (d9)dihydroceramide is 549.5 → 266.3 $m/z$ (declustering potential, 130 V; collision energy, 38 V). The MS instrument was operated in positive ion mode (curtain gas, 35 arbitrary units (AU); collision gas, medium; ion spray voltage, 4,200 V; temperature, 550 °C; ion source gas 1, 65 AU; ion source gas 2, 80 AU). MS data processing was performed in Gubbs Mass Spec Utilities and peak area ratios between C16:0 dihydroceramide and the internal standard were calculated.

Expression of each mutant in the membrane samples was evaluated through western blotting using 10 ng ml$^{-1}$ Strep-Tactin conjugated with horseradish peroxidase (IBA Lifesciences), applying SYPRO ruby (Thermo Fisher Scientific) staining as a loading control (Fig. 3e). Band intensity was quantified using the GeneTools software (Syngene) and activity was normalized to protein expression.

## MD simulations

Atomistic MD simulations were performed using the Desmond software package (D. E. Shaw Research) within the Maestro software suite (Schrödinger). The simulation setup and analysis were carried out as follows. For each of the two states, a CerS6 monomer was embedded in a lipid bilayer consisting of 99 POPC molecules after undergoing the protein preparation step as implemented in Maestro. The dimensions of the simulation cell were approximately 90 × 70 × 70 Å, with a minimum distance of 10 Å between the protein and the cell boundaries. The systems contained 8,855 (covalent intermediate bound state) or 9,368 ($N$-acyl FB$_1$-bound state) water molecules and eight or six corresponding chloride counterions to maintain overall charge neutrality. The simulations were performed using the OPLS4 force field[70] for the protein and lipid molecules and the simple point-charge water model for the water molecules. The system was set up using the Maestro software suite (Schrödinger). The simulations were run in Desmond using an NPT ensemble at a temperature of 300 K. A timestep of 2 fs was used for the integration of equations of motion. Four independent simulations were performed, each with a duration of 100 ns, resulting in a total simulation time of 400 ns. Snapshots of the system were recorded every 100 ps for further analysis.

## Figures

Figures depicting molecular models were generated using PyMOL (Schrödinger) and ChimeraX[71].

## Reporting summary

Further information on research design is available in the Nature Portfolio Reporting Summary linked to this article.

## Data availability

The cryo-EM maps were deposited to the EM Data Bank under accession codes EMD-18770 (CerS6–Nb22 covalent acyl–enzyme intermediate state), EMD-18771 (CerS6–Nb22 $N$-acyl FB$_1$-bound state) and

EMD-19869 (CerS6–Nb02 covalent acyl–enzyme intermediate state). The atomic models were deposited to the PDB under accession codes 8QZ6 (CerS6–Nb22 covalent acyl–enzyme intermediate state), 8QZ7 (CerS6–Nb22 $N$-acyl $FB_1$-bound state) and 9EOT (CerS6–Nb02 covalent acyl–enzyme intermediate state). All raw MS data are available for download from Zenodo (https://doi.org/10.5281/zenodo.10604228)[72]. Source data are provided with this paper.

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

## Acknowledgements

T.C.P. was supported by a Wellcome PhD studentship (102164/B/15/Z). Initial work on this project was funded by the Structural Genomics Consortium, a registered charity (number 1097737) that received funds from AbbVie, Bayer, Boehringer Ingelheim, the Canada Foundation for Innovation, Genome Canada, Janssen, Merck, Novartis, the Ontario Ministry of Economic Development and Innovation, Pfizer and Takeda, as well as the Wellcome Trust (106169/Z/14/Z). D.B.S., A.C.W.P. and G.C. were supported by the Innovative Medicines Initiative 2 Joint Undertaking (JU) under grant agreement no. 875510. The JU receives support from the European Union's Horizon 2020 research and innovation program, the European Federation of Pharmaceutical Industries and Associations, the Ontario Institute for Cancer Research, the Royal Institution for the Advancement of Learning McGill University, Kungliga Tekniska Hoegskolan and Diamond Light Source, Ltd. EM was provided through OPIC, an Instruct European Research Infrastructure Consortium center (funded by Wellcome Trust Joint Infrastructure Fund award (060208/Z/00/Z) and equipment grant (093305/Z/10/Z)) and the Midlands Regional Cryo-EM Facility at the LISCB, with major funding from the Medical Research Council (MC_PC_17136). We thank C. S. Savva (LISCB) for technical assistance with cryo-EM data collection and W. Greenland (Agilent) for technical advice on MS/MS. We thank W. Yue and C. Siebold for helpful discussions.

## Author contributions

T.C.P. expressed and purified protein samples, prepared proteoliposome samples for alpaca immunization and membranes for mutant activity testing, carried out biophysical characterizations including BLI experiments, designed and generated CerS6 mutants through site-directed mutagenesis, built and refined the atomic models and performed intact protein MS, product identification by HRMS and small-molecule structural characterization by LC–ESI-MS/MS. T.C.P. and G.C. prepared cryo-EM grids. T.C.P. and A.C.W.P. collected and processed the cryo-EM data. C.S.T. carried out the MD simulations and analysis. A.Q. was involved in the early stages of the project, including initial screening of expression and purification conditions. R.C. provided access to MS instruments for intact protein and small-molecule analysis and assisted with small-molecule MS experiments. S.Š. conducted alpaca immunizations, nanobody library generation, panning and identification of unique nanobody sequences. M.T. performed the CerS6 mutant activity assays, supervised by S.T. T.C.P. and D.B.S. wrote the original draft of the paper. All authors participated in the discussion and paper editing. A.P., E.P.C., G.S. and D.B.S. supervised the research.

## Competing interests

C.S.T., M.T., S.T., A.P. and G.S. are employees of Boehringer Ingelheim Pharma, GmbH & Co. KG. The other authors declare no competing interests.

## Additional information

**Extended data** is available for this paper at https://doi.org/10.1038/s41594-024-01414-3.

**Correspondence and requests for materials** should be addressed to Tomas C. Pascoa, Elisabeth P. Carpenter, Gisela Schnapp or David B. Sauer.

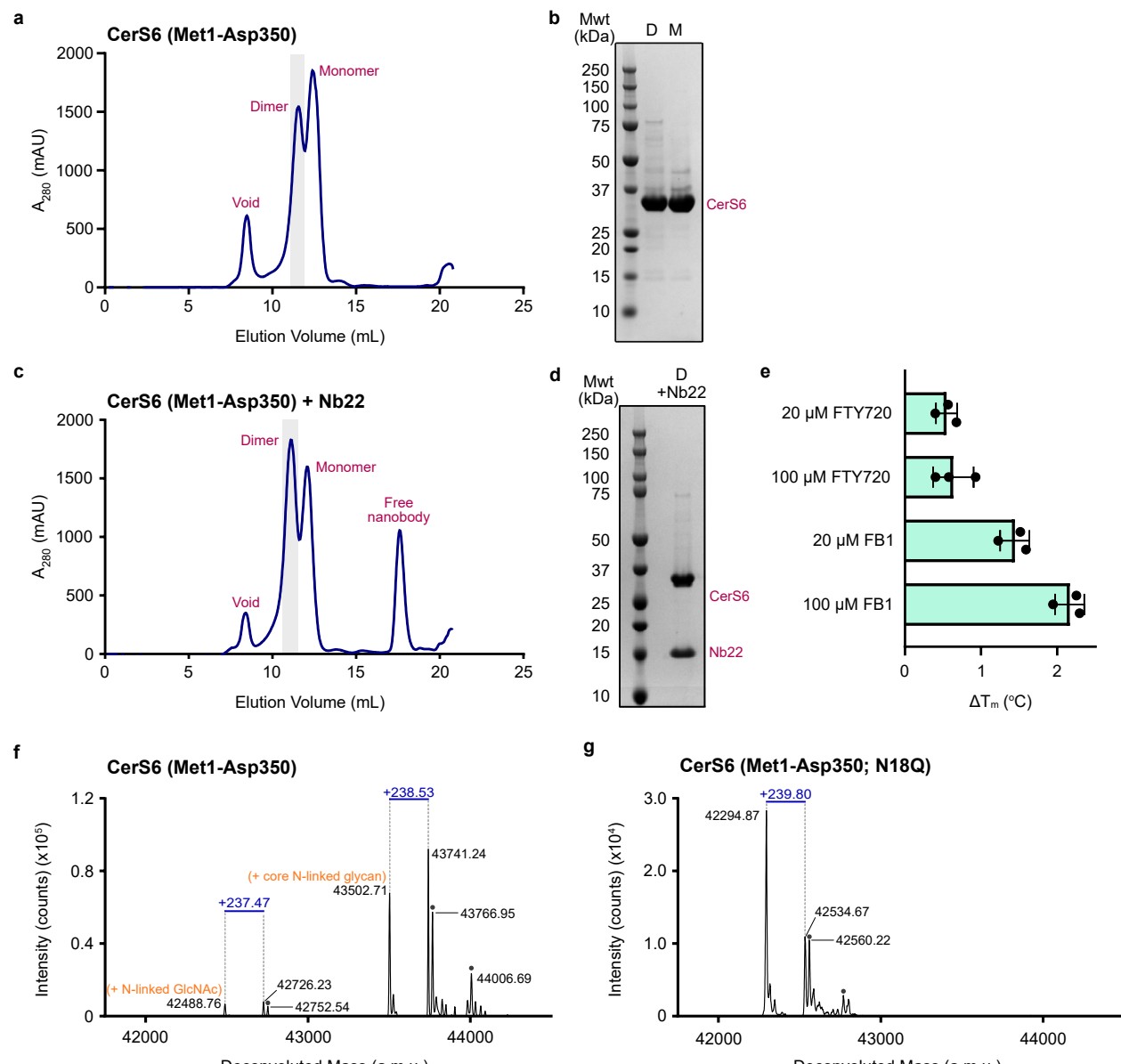

**Extended Data Fig. 1 | Properties of purified CerS6. a**, Elution profile of CerS6 (Met1-Asp350). The collected dimer peak is shaded in gray. **b**, SDS-PAGE analysis of the purified dimeric (D) and monomeric (M) SEC peak fractions. Similar results were obtained for all purifications tested (n = 6). **c**, Elution profile of CerS6 (Met1-Asp350) in complex with nanobody 22. The putative dimer peak, shaded in gray, was pooled and concentrated for single particle cryo-EM. **d**, SDS-PAGE analysis of the SEC-purified CerS6 dimer and nanobody 22 complex. Similar results were obtained for all purifications tested (n = 3). **e**, NanoDSF screening of CerS inhibitors. n = 3 technical replicates. Error bars show the SEM. **f-g**, Denaturing intact protein MS analysis of purified CerS6. **f**, Deconvoluted mass spectra of wild-type CerS6 (Met1-Asp350). The expected mass of the untagged, unmodified, truncated enzyme based on the sequence is 42,373.53 Da. The observed lower mass peak (42,488.76 Da) corresponds to loss of the initiator methionine (−131.20 Da), acetylation of the new N-terminus ( + 42.04 Da), and addition of an

N-linked GlcNAc ( + 203.19 Da). An additional, higher intensity, deconvoluted mass peak (43,502.71 Da) corresponds to the addition of a core N-linked glycan (theoretical +1217.09 Da) in place of simply the N-linked GlcNAc. Mass shifts corresponding to the mass of a palmitoyl group (theoretical +238.41 Da) are labelled in blue. Additional deconvoluted mass peaks corresponding to the addition of an unknown modification of approximately +264 Da on top of the glycosylation or glycosylation + palmitoylation peaks are labelled with gray dots. **g**, Deconvoluted mass spectra obtained for the purified CerS6 N18Q mutant which removes the only identified glycosylation site. The expected mass of the untagged, unmodified, truncated CerS6 N18Q enzyme based on the sequence is 42,387.56 Da. The observed lower mass peak (42,294.87 Da) corresponds to loss of the initiator methionine followed by acetylation of the new N-terminus. The +239.80 Da mass shift, corresponding to palmitoylated protein, is labelled in blue.

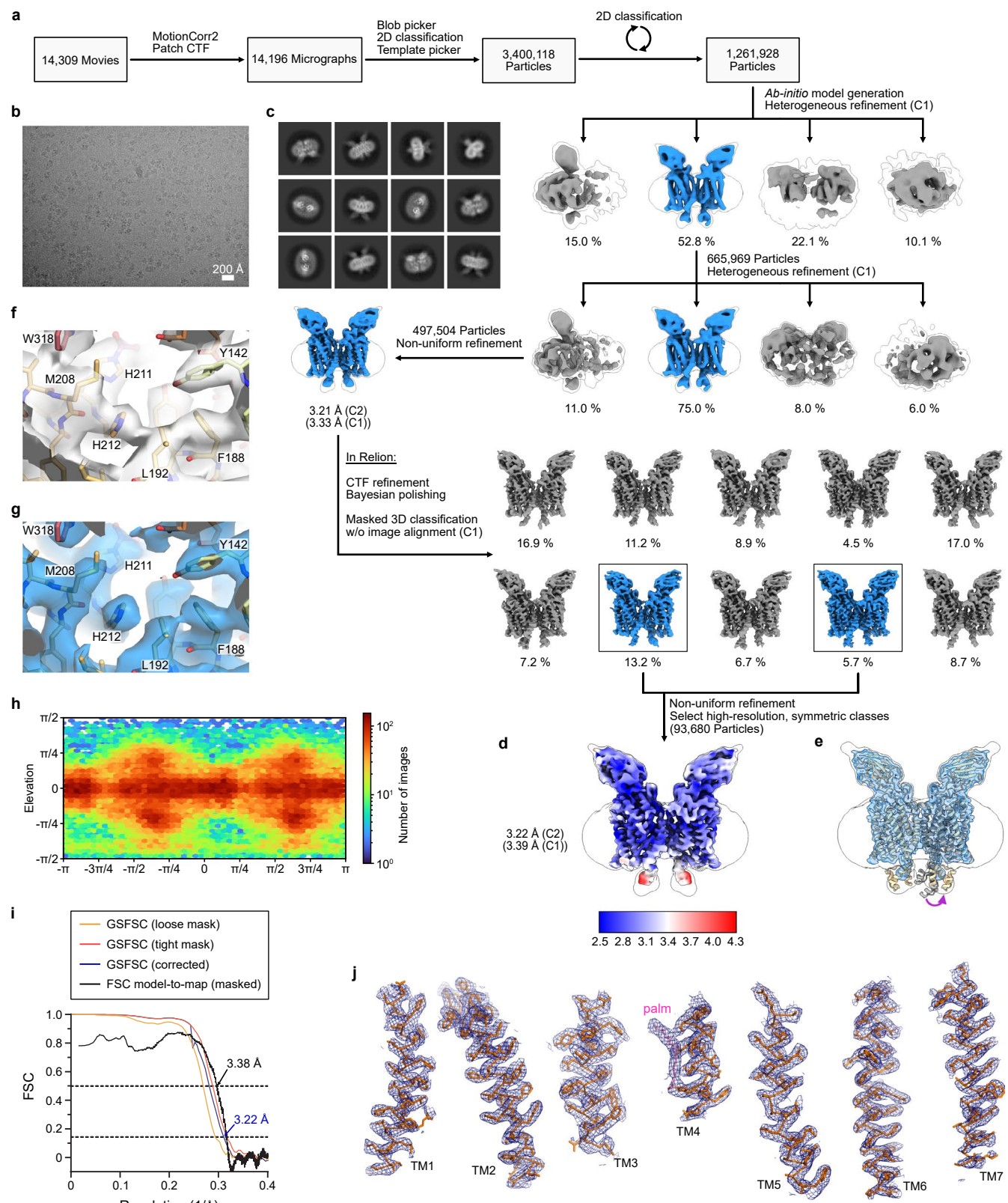

**Extended Data Fig. 2 | Cryo-EM data processing: CerS6-Nb22 covalent intermediate dataset. a**, Cryo-EM data processing flowchart. **b**, Representative micrograph. **c**, 2D classes obtained prior to 3D classification. **d**, Final 3D reconstruction, colored by local resolution. **e**, Overlay of the structural model (tan) and the AlphaFold2 monomer prediction (gray). The cryo-EM maps are shown as blue (unsharpened map) or outline (blurred map; $B_{blur}$ = 200 Å²) surfaces. A purple arrow indicates the manual adjustment of the position of the

Hox-like domain into the experimental blurred map. **f-g**, Cryo-EM map **(f)** before and **(g)** after CTF refinement, Bayesian polishing and 3D classification in Relion. **h**, Angular distribution of particles used in the final reconstruction. **i**, Fourier Shell Correlation (FSC) plots, indicating overall map resolution (GSFSC = 0.143) and a model-to-map FSC curve. **j**, Sharpened cryo-EM map (contoured at 4.6σ) overlaid on the final model.

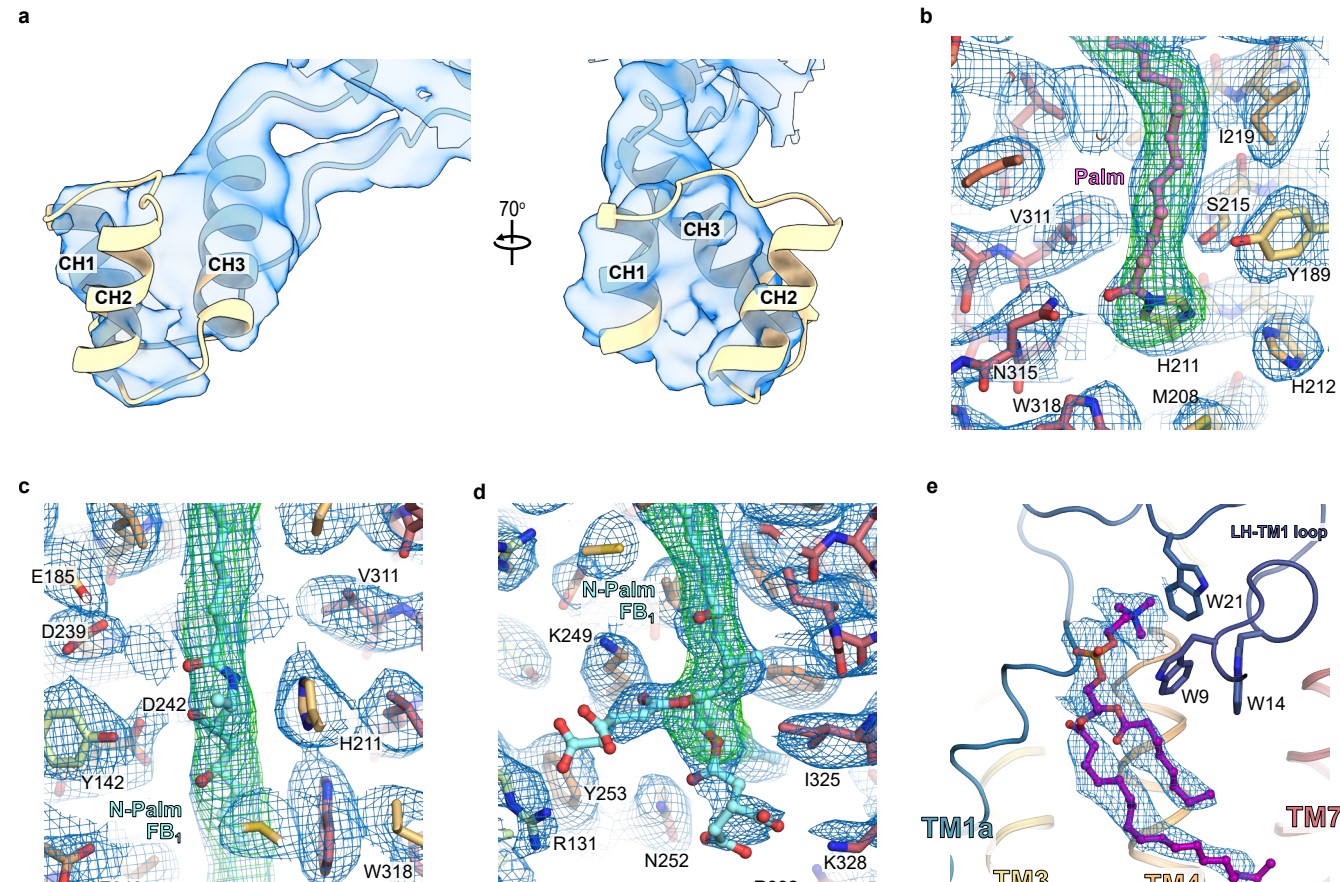

**Extended Data Fig. 3 | Cryo-EM map quality in the CerS6-Nb22 datasets. a**, Fit of the Hox-like domain into the unsharpened cryo-EM map (blue surface). **b-d**, Cryo-EM density in the regions around the modelled ligands. The respective sharpened cryo-EM maps (blue mesh) and Servalcat Fo-Fc difference maps (obtained by omitting the acyl-imidazole and the N-palmitoyl fumonisin $B_1$ species from the models; green mesh) are overlaid on the models. **e**, Cryo-EM density for the bound lipid molecule, modelled as phosphatidylcholine (purple carbon atoms).

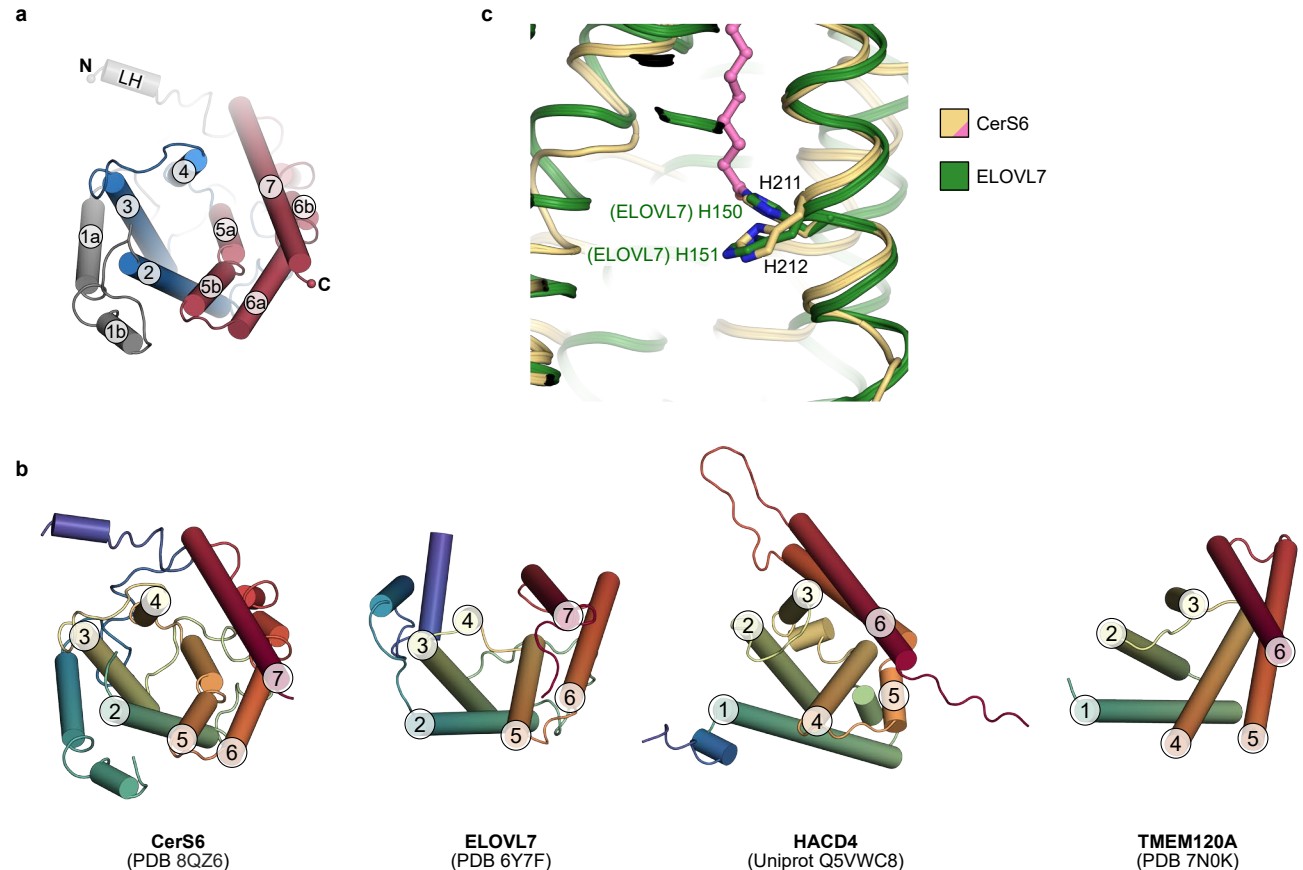

**Extended Data Fig. 4 | Transmembrane helix topology of CerS6. a**, The 6-TM barrel formed by TM2-7 of CerS6 is composed of two 3-TM units (TM2-4, blue; TM5-7, red), arranged as inverted repeats. **b**, Comparison of the transmembrane helix topology of the 6-TM barrels of CerS6, ELOVL7, HACD4 and TMEM120A (TACAN). **c**, Structural alignment of CerS6 and ELOVL7 reveals that their histidine pairs are structurally homologous. The acyl chain linked to His211 in the CerS6 covalent intermediate structure is shown in pink.

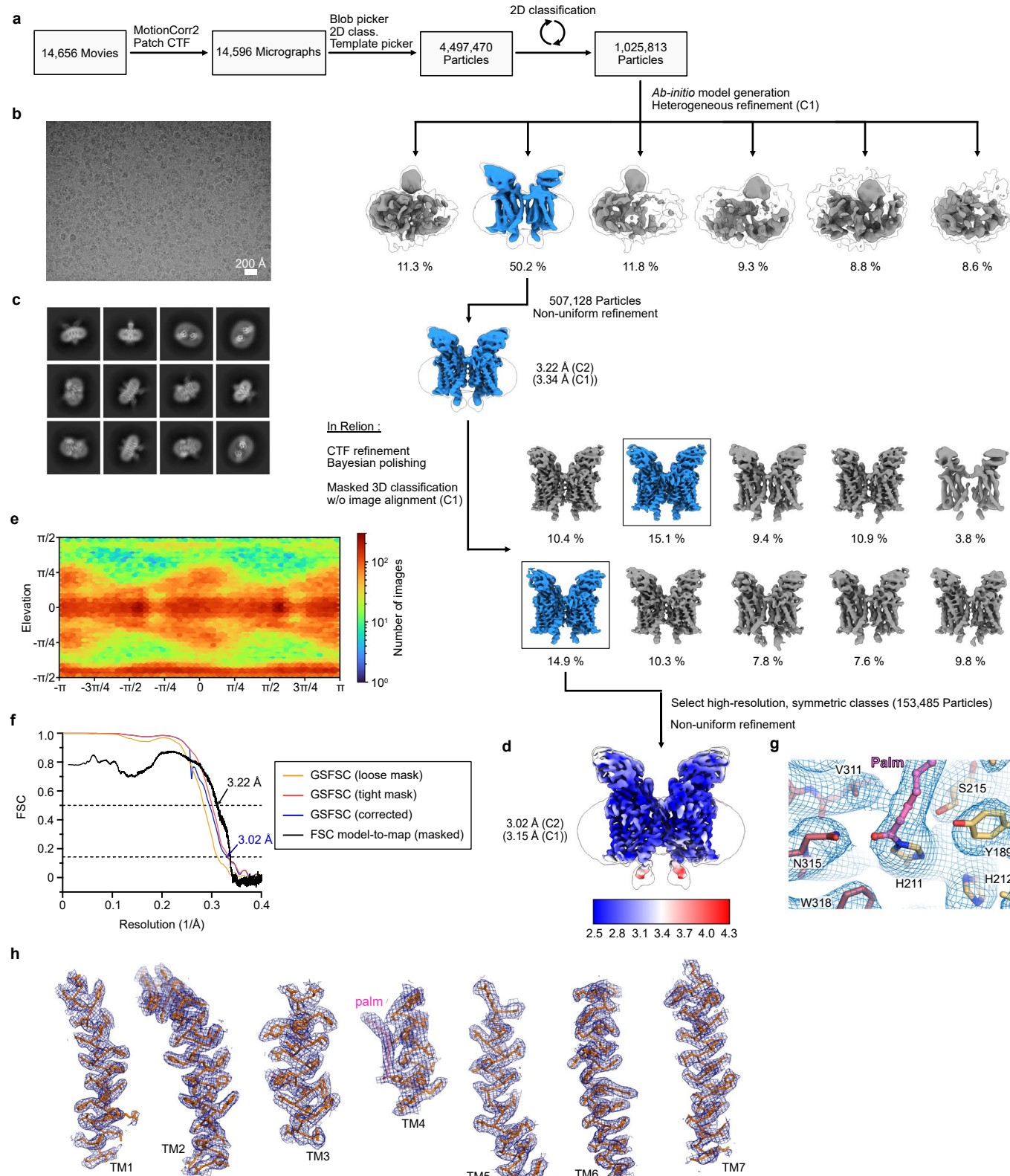

**Extended Data Fig. 5 | Cryo-EM data processing: CerS6-Nb02 covalent intermediate dataset. a**, Cryo-EM data processing flowchart. **b**, Representative micrograph. **c**, 2D classes obtained prior to 3D classification. **d**, Final 3D reconstruction, colored by local resolution. **e**, Angular distribution of particles used in the final reconstruction. **f**, Fourier Shell Correlation (FSC) plots, indicating overall map resolution (GSFSC = 0.143) and a model-to-map FSC curve. **g**, Coulombic potential map in the region around the covalent linkage of the acyl chain to His211. **h**, Sharpened cryo-EM map (contoured at 4.6σ) overlaid on the final model.

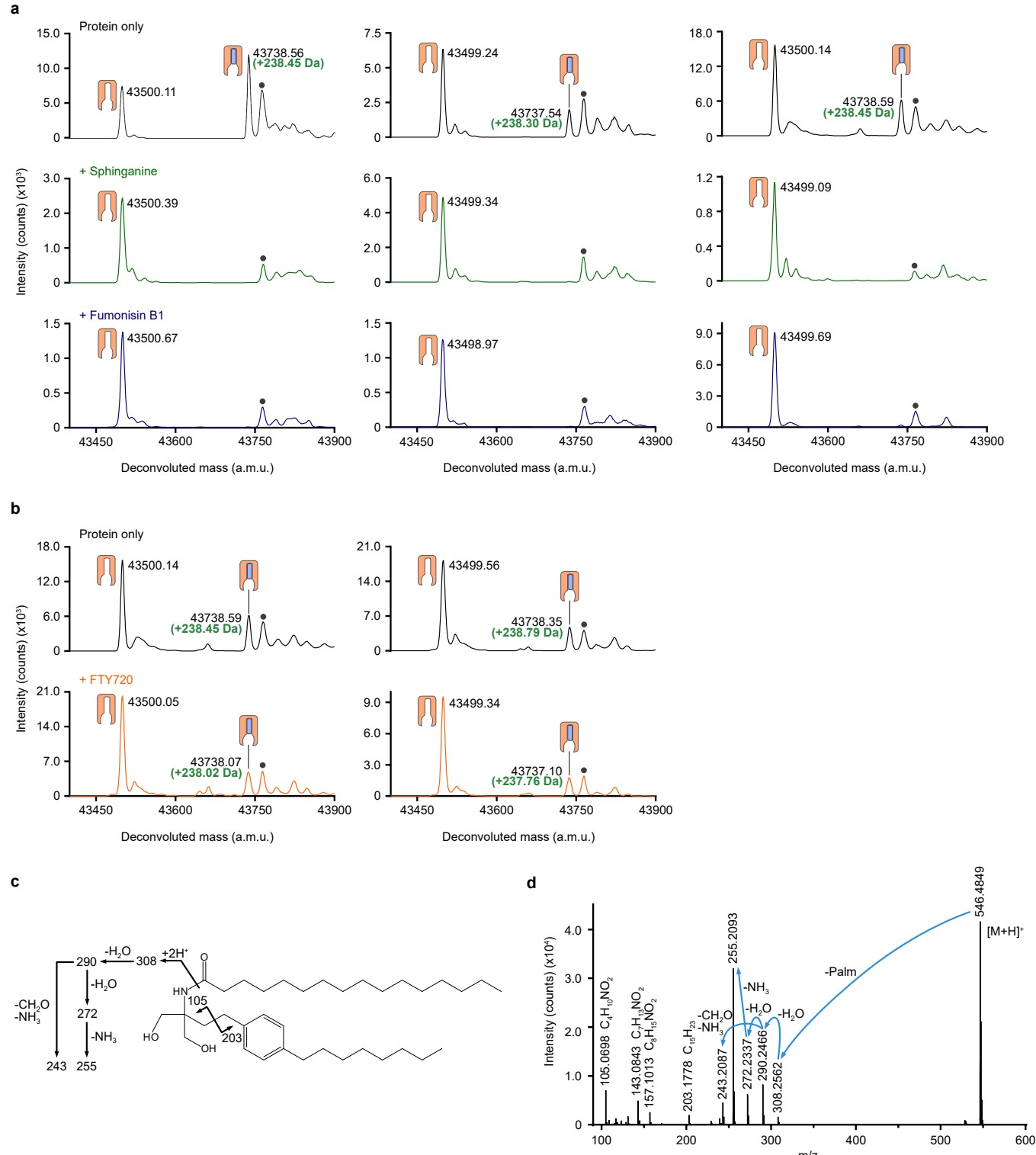

**Extended Data Fig. 6 | Monitoring the reaction of the covalent acyl-enzyme intermediate by mass spectrometry. a-b**, Purified CerS6 was incubated in the presence and absence of substrates prior to liquid chromatography – electrospray ionization – mass spectrometry (LC-ESI-MS) intact mass analysis. **a**, Deconvoluted mass spectra are shown for CerS6 incubated in the absence of substrates (protein only), with 200 μM sphinganine, or with 200 μM fumonisin B₁. Deconvoluted mass peaks are indicated as follows: unmodified enzyme, orange icon; covalent acyl-enzyme species, orange and pink icon; background species ( + 264 Da) present in all traces, gray circle. Spectra are

shown for 3 biological replicates. **b**, Deconvoluted mass spectra shown for CerS6 incubated in the presence and absence of 600 μM FTY720. Spectra are shown for 2 biological replicates. **c-d** Structural characterization of N-palmitoyl FTY720 species by LC-ESI-MS/MS. **c**, Structure of the proposed N-palmitoyl FTY720 reaction product, annotated with the m/z values of the daughter ions observed in panel d. **d**, MS/MS spectrum obtained from the fragmentation of the [M + H]⁺ ion of N-palmitoyl FTY720 (m/z 546.4881) using a collision-induced dissociation energy of 14 V. Proposed relationships between the observed m/z peaks are indicated.

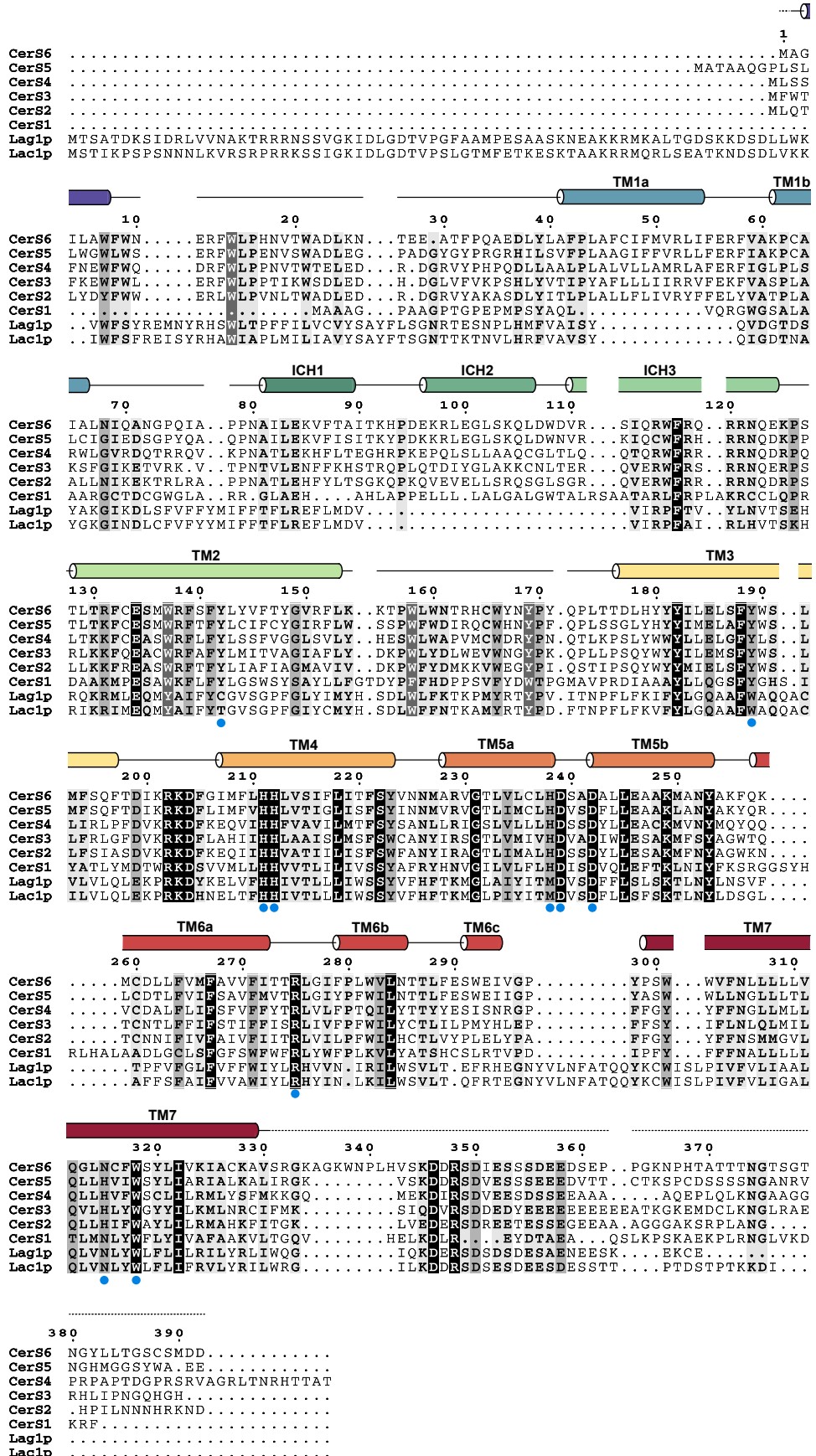

**Extended Data Fig. 7 | Multiple sequence alignment of human (CerS1-6) and S. cerevisiae (Lag1p, Lac1p) ceramide synthases.** Blue circles below the alignment indicate the active site residues.

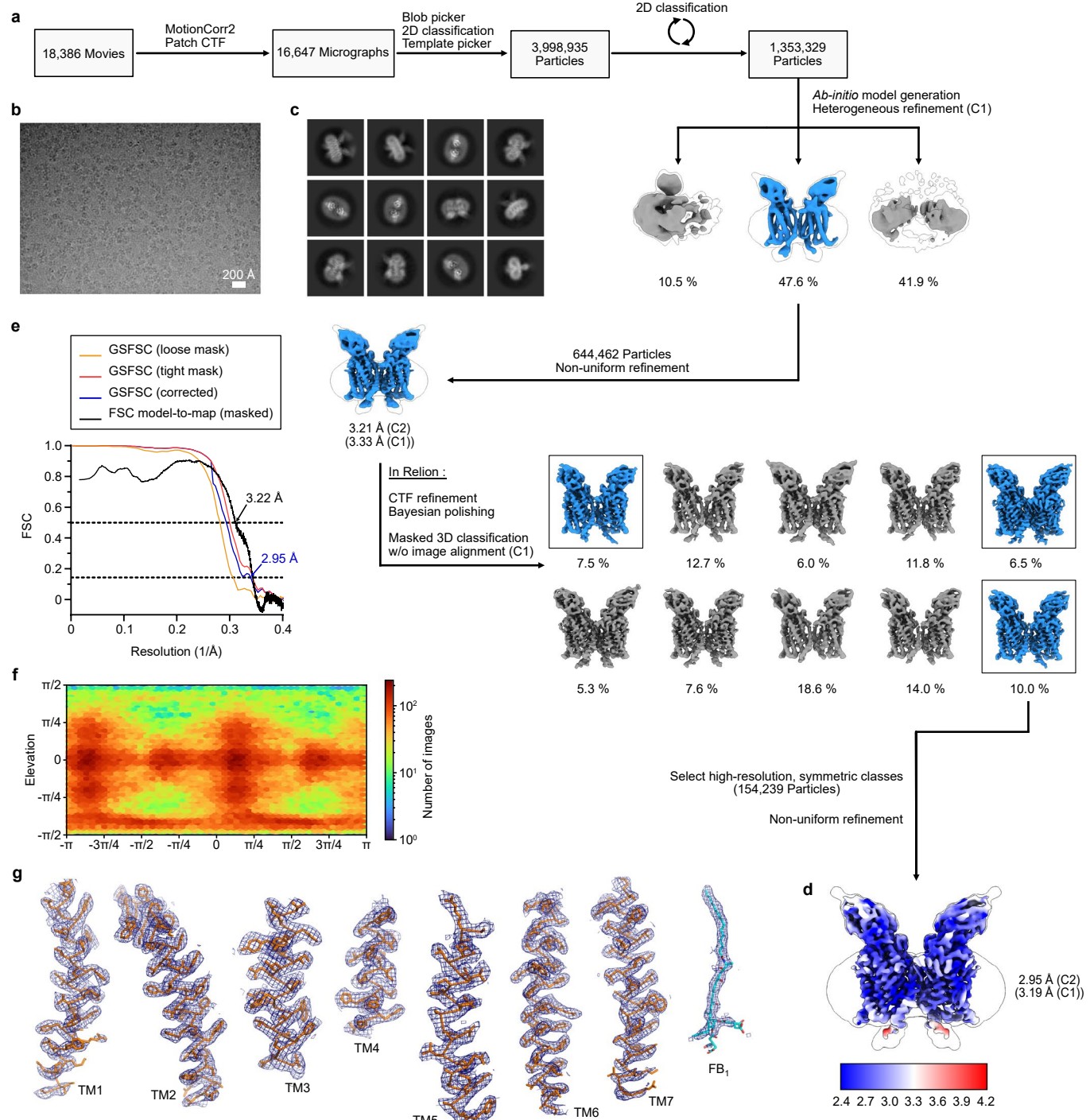

**Extended Data Fig. 8 | Cryo-EM data processing: CerS6-Nb22 N-palmitoyl FB₁-bound dataset. a**, Cryo-EM data processing flowchart. **b**, Representative micrograph. **c**, 2D classes obtained prior to 3D classification. **d**, Final 3D reconstruction, colored by local resolution. **e**, Fourier Shell Correlation (FSC) plots, indicating overall map resolution (GSFSC = 0.143) and a model-to-map FSC curve. **f**, Angular distribution of particles used in the final reconstruction. **g**, Sharpened cryo-EM map (contoured at 4.6σ) overlaid on the final model.

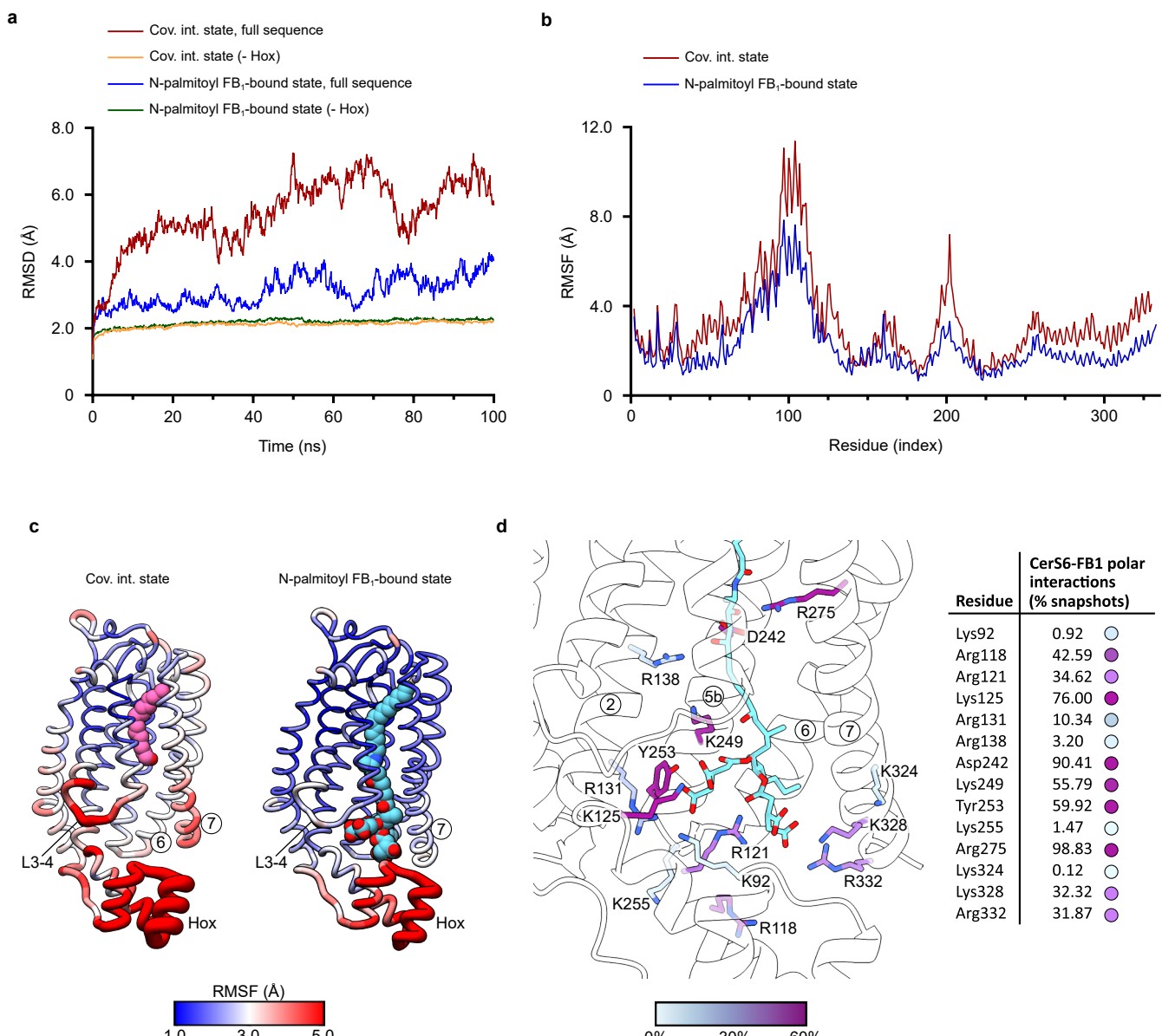

**Extended Data Fig. 9 | CerS6 dynamics and N-palmitoyl fumonisin B$_1$ interactions in simulations. a**, Average root-mean-square deviation (RMSD) values of Cα atoms from atomistic simulations of the covalent intermediate and N-palmitoyl fumonisin B$_1$-bound states. Traces where the Hox-like domain residues (70-126) were excluded from the RMSD calculation are shown to highlight the stability of the transmembrane region. **b**, Root-mean-square fluctuation (RMSF) values by residue (Cα atoms). RMSD and RMSF values shown are averaged over 4 ×100 ns atomistic simulations. **c**, N-palmitoyl FB$_1$

stabilises the CerS6 structure. RMSF values from the simulations are mapped onto the structures, revealing a reduction in the overall dynamics of the protein upon binding the reaction product N-palmitoyl FB$_1$, particularly in the Hox-like domain, the TM3-4 loop, TM6, and TM7. **d**, Analysis of the polar interactions between CerS6 and FB$_1$ during the simulations. Values shown correspond to the percentage of snapshots (taken every 100 ps during the simulations) containing CerS6-FB$_1$ polar interactions at each residue. Residues are coloured by the frequency of polar interactions with FB$_1$ (cyan sticks).

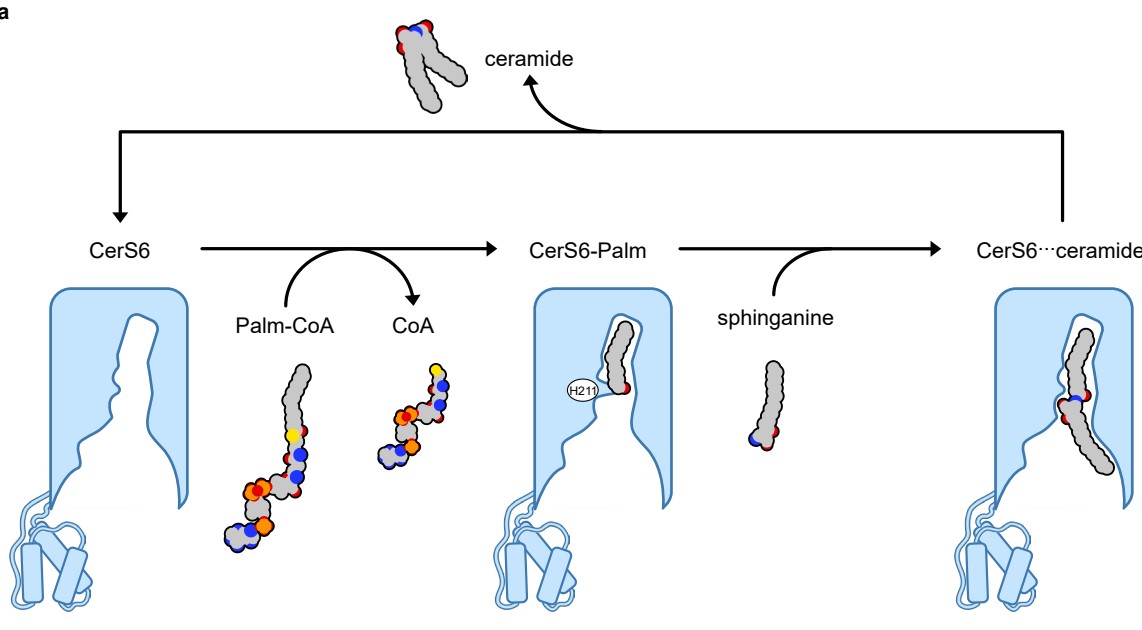

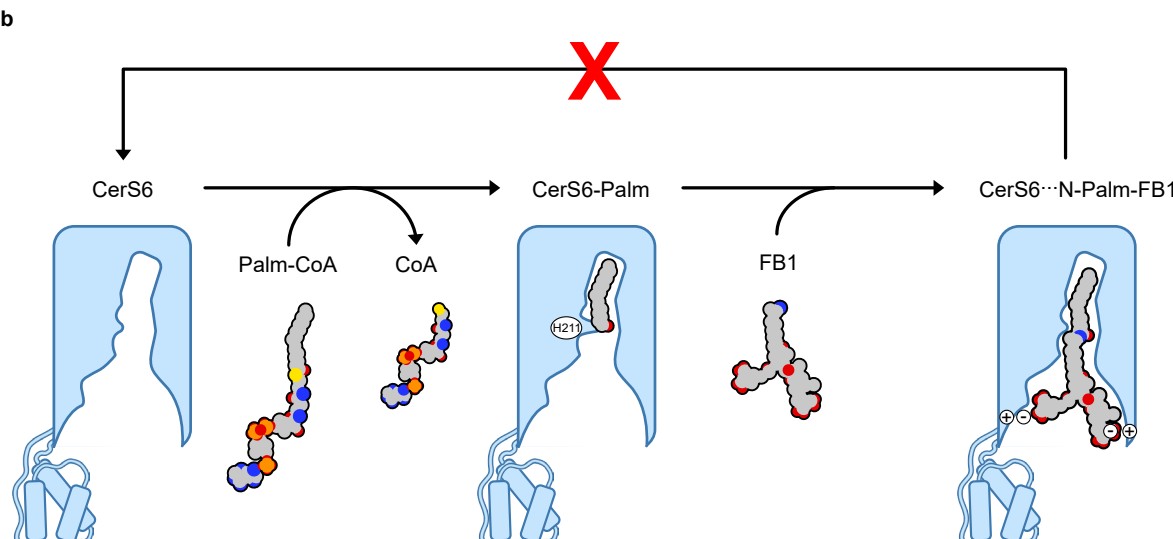

**Extended Data Fig. 10 | Cartoon representation of the proposed mechanism of ceramide synthases and their inhibition by FB₁. a**, CerS6 initially reacts with an acyl-CoA substrate, forming a covalent acyl-enzyme intermediate and releasing CoA as the first product. Subsequently, the reaction of the acyl-enzyme intermediate with sphinganine produces ceramide as the final product. Ceramide dissociates from the enzyme, which can then react with another acyl-CoA substrate to re-start the cycle. **b**, Reaction of the covalent acyl-enzyme intermediate species with FB₁ results in the formation of an N-acyl FB₁ product. This inhibitory product remains tightly bound in the central cavity, forming polar interactions near the cytoplasmic entrance, and preventing recycling of the enzyme.

# Reporting Summary

## Statistics

For all statistical analyses, confirm that the following items are present in the figure legend, table legend, main text, or Methods section.

| n/a | Confirmed | |
|---|---|---|
| ☐ | ☒ | The exact sample size (*n*) for each experimental group/condition, given as a discrete number and unit of measurement |
| ☐ | ☒ | A statement on whether measurements were taken from distinct samples or whether the same sample was measured repeatedly |
| ☒ | ☐ | The statistical test(s) used AND whether they are one- or two-sided *Only common tests should be described solely by name; describe more complex techniques in the Methods section.* |
| ☒ | ☐ | A description of all covariates tested |
| ☒ | ☐ | A description of any assumptions or corrections, such as tests of normality and adjustment for multiple comparisons |
| ☒ | ☐ | A full description of the statistical parameters including central tendency (e.g. means) or other basic estimates (e.g. regression coefficient) AND variation (e.g. standard deviation) or associated estimates of uncertainty (e.g. confidence intervals) |
| ☒ | ☐ | For null hypothesis testing, the test statistic (e.g. *F*, *t*, *r*) with confidence intervals, effect sizes, degrees of freedom and *P* value noted *Give P values as exact values whenever suitable.* |
| ☒ | ☐ | For Bayesian analysis, information on the choice of priors and Markov chain Monte Carlo settings |
| ☒ | ☐ | For hierarchical and complex designs, identification of the appropriate level for tests and full reporting of outcomes |
| ☒ | ☐ | Estimates of effect sizes (e.g. Cohen's *d*, Pearson's *r*), indicating how they were calculated |

*Our web collection on statistics for biologists contains articles on many of the points above.*

## Software and code

Policy information about availability of computer code

| Data collection | Cryo-EM data were collected with EPU (version 2.13; Thermo Fisher Scientific). |
|---|---|
| Data analysis | Cryo-EM data were processed and analyzed with cryoSPARC (version 3.3.1; Structura Biotechnology) and RELION (version 3.1.3). Additional software from the Collaborative Computational Projects No. 4 (CCP4; v7.0) and  Electron cryo-Microscopy (CCPEM; v1.6.0) software suites were also used. Structural models were built using Coot (v0.8.9.5) with model refinement performed by PHENIX (v1.20). Molecular visualisation, analysis and figure generation were performed using PyMOL 2.2.0 (Schrodiner) and ChimeraX 1.16 (UCSF Resource for Biocomputing, Visualization, and Informatics). Intact protein mass spectrometry data was analyzed using MassHunter Qualitative Analysis v.B.07.00 (Agilent Technologies) software. Molecular dynamics simulations were performed using Desmond 7.4.121 as provided in the Maestro suite 2023-2. |

For manuscripts utilizing custom algorithms or software that are central to the research but not yet described in published literature, software must be made available to editors and reviewers. We strongly encourage code deposition in a community repository (e.g. GitHub). See the Nature Portfolio guidelines for submitting code & software for further information.

## Data

Policy information about availability of data

All manuscripts must include a data availability statement. This statement should provide the following information, where applicable:
- Accession codes, unique identifiers, or web links for publicly available datasets
- A description of any restrictions on data availability
- For clinical datasets or third party data, please ensure that the statement adheres to our policy

The cryo-EM maps have been deposited in the Electron Microscopy Data Bank (EMDB) under accession codes EMD-18770 (CerS6-Nb22 covalent acyl-enzyme intermediate state), EMD-18771 (CerS6-Nb22 N-acyl FB1-bound state) and EMD-19869 (CerS6-Nb02 covalent acyl-enzyme intermediate state). The atomic models have been deposited in the Protein Data Bank under accession codes 8QZ6 (CerS6-Nb22 covalent acyl-enzyme intermediate state), 8QZ7 (CerS6-Nb22 N-acyl FB1-bound state) and 9EOT (CerS6-Nb02 covalent acyl-enzyme intermediate state). All the raw mass spectrometry data are available for download on the Zenodo repository under DOI: 10.5281/zenodo.10604228.

## Research involving human participants, their data, or biological material

Policy information about studies with human participants or human data. See also policy information about sex, gender (identity/presentation), and sexual orientation and race, ethnicity and racism.

| | |
|---|---|
| Reporting on sex and gender | N/A |
| Reporting on race, ethnicity, or other socially relevant groupings | N/A |
| Population characteristics | N/A |
| Recruitment | N/A |
| Ethics oversight | N/A |

Note that full information on the approval of the study protocol must also be provided in the manuscript.

# Field-specific reporting

Please select the one below that is the best fit for your research. If you are not sure, read the appropriate sections before making your selection.

☒ Life sciences    ☐ Behavioural & social sciences    ☐ Ecological, evolutionary & environmental sciences

For a reference copy of the document with all sections, see nature.com/documents/nr-reporting-summary-flat.pdf

# Life sciences study design

All studies must disclose on these points even when the disclosure is negative.

| | |
|---|---|
| Sample size | No statistical method was used to predetermine sample size. Study samples sizes were chosen to ensure reproducibility based on prior experimental experience, including Nie et al. NSMB 2021 |
| Data exclusions | None |
| Replication | The dihydroceramide synthase assays were performed on 3 biological replicates. For each biological replicate, 4 technical replicates were used. |

For the intact protein mass spectrometry data, we present biological replicates as an Extended Data Figure. All experiments were repeated at least twice using different protein samples purified independently and MS analysis was carried out on different days. Replicates gave similar results. Cryo-EM data collection was only carried out on a single sample in each case.

Randomization    For cryo-EM studies, particles were randomly assigned to half-maps for resolution determination. No randomization was not used for dihydroceramide synthase or intact protein MS assays.

Blinding    Blinding was not used. It is not relevant as outcomes of the cryo-EM or mass spectrometry experiments are not affected by knowledge of the sample.

# Reporting for specific materials, systems and methods

We require information from authors about some types of materials, experimental systems and methods used in many studies. Here, indicate whether each material, system or method listed is relevant to your study. If you are not sure if a list item applies to your research, read the appropriate section before selecting a response.

| Materials & experimental systems | Methods |
|---|---|
| n/a | Involved in the study | n/a | Involved in the study |
| ☒ ☐ Antibodies | ☒ ☐ ChIP-seq |
| ☐ ☒ Eukaryotic cell lines | ☒ ☐ Flow cytometry |
| ☒ ☐ Palaeontology and archaeology | ☒ ☐ MRI-based neuroimaging |
| ☐ ☒ Animals and other organisms | |
| ☒ ☐ Clinical data | |
| ☒ ☐ Dual use research of concern | |
| ☒ ☐ Plants | |

## Eukaryotic cell lines

Policy information about cell lines and Sex and Gender in Research

Cell line source(s)    Spodoptera frugiperda (Sf9) cells (Thermo-Fisher Scientific, Cat. No. 11496015). Expi293F GnTI- cells (Cat# A39240; Thermo Fisher Scientific)

Authentication    Cell lines used (SF9/Expi293F) are standard laboratory model overexpression strains purchased from Thermo Fisher. These cell lines undergo quality control before dispatch. Cells were passaged a limited number of times before a new batch from the manufacturer was employed. Cells were monitored by regular visual inspection.

Mycoplasma contamination    Not performed

Commonly misidentified lines    No commonly misidentified cell lines were used in this study.
(See ICLAC register)

## Animals and other research organisms

Policy information about studies involving animals; ARRIVE guidelines recommended for reporting animal research, and Sex and Gender in Research

Laboratory animals    Adult Alpaca (Vicugna pacos) named Vesus 4.5 years old  at the time of immunization

Wild animals    No wild animals were used in this study.

Reporting on sex    Male alpaca was immunized, though sex is not relevant to results as the alpaca was used only for nanobody  generation.

Field-collected samples    No field collected samples were used in this study.

Ethics oversight    The immunizations of alpaca were conducted strictly according to the guidelines of the Swiss Animals Protection Law and were approved by the Cantonal Veterinary Office of Zurich, Switzerland (Licenses No. ZH 198/17 and ZH028/2021).

Note that full information on the approval of the study protocol must also be provided in the manuscript.

## Plants

Seed stocks

*Report on the source of all seed stocks or other plant material used. If applicable, state the seed stock centre and catalogue number. If plant specimens were collected from the field, describe the collection location, date and sampling procedures.*

Novel plant genotypes

*Describe the methods by which all novel plant genotypes were produced. This includes those generated by transgenic approaches, gene editing, chemical/radiation-based mutagenesis and hybridization. For transgenic lines, describe the transformation method, the number of independent lines analyzed and the generation upon which experiments were performed. For gene-edited lines, describe the editor used, the endogenous sequence targeted for editing, the targeting guide RNA sequence (if applicable) and how the editor was applied.*

Authentication

*Describe any authentication procedures for each seed stock used or novel genotype generated. Describe any experiments used to assess the effect of a mutation and, where applicable, how potential secondary effects (e.g. second site T-DNA insertions, mosiacism, off-target gene editing) were examined.*

