## [Peer Review File · Nature Structural & Molecular Biology]

Structural basis of the mechanism and inhibition of a human ceramide synthase

Corresponding Author: Dr David Sauer

Version 0:

Decision Letter:

11th Jan 2024

Dear Dr. Sauer,

Thank you again for submitting your manuscript "Structural basis of the mechanism and inhibition of a human ceramide synthase". We now have comments (below) from the 2 reviewers who evaluated your paper. In light of those reports, we remain interested in your study and would like to see your response to the comments of the referees, in the form of a revised manuscript.

You will see that while reviewers appreciate the results, they raise several concerns which will need to be addressed in a revision.

Specifically, please make an effort to fully address concerns of reviewer #1 pertaining to mass spectrometry data reporting, analysis and interpretation, as well as model building of the lipid molecules in the structure. Please also revisit the interpretation of the mutagenesis experiments in respect to the proposed model, taking into account suggestions of reviewer #1. While we agree with reviewer #1 that enzyme kinetic analysis would further support the ping-pong mechanism, we do not consider it essential in the context of the current work.

Please be sure to address/respond to all concerns of the referees in full in a point-by-point response and highlight all changes in the revised manuscript text file. If you have comments that are intended for editors only, please include those in a separate cover letter.

We expect to see your revised manuscript within 12 weeks. If you cannot send it within this time, please contact us to discuss an extension; we would still consider your revision, provided that no similar work has been accepted for publication at NSMB or published elsewhere.

Reporting Summary:

-- that unprocessed scans are clearly labelled and match the gels and western blots presented in figures.

-- that control panels for gels and western blots are appropriately described as loading on sample processing controls
-- all images in the paper are checked for duplication of panels and for splicing of gel lanes.

Please note that all key data shown in the main figures as cropped gels or blots should be presented in uncropped form, with molecular weight markers. These data can be aggregated into a single supplementary figure item. While these data can be displayed in a relatively informal style, they must refer back to the relevant figures. These data should be submitted with the final revision, as source data, prior to acceptance, but you may want to start putting it together at this point.

Data availability: this journal strongly supports public availability of data. All data used in accepted papers should be available via a public data repository, or alternatively, as Supplementary Information. If data can only be shared on request, please explain why in your Data Availability Statement, and also in the correspondence with your editor. Please note that for some data types, deposition in a public repository is mandatory - more information on our data deposition policies and available repositories can be found below:

<https://www.nature.com/nature-research/editorial-policies/reporting-standards#availability-of-data>

Link Redacted

Sincerely,

Katarzyna Ciazynska, PhD
(she/her)
Associate Editor
Nature Structural & Molecular Biology
<https://orcid.org/0000-0002-9899-2428>

Referee expertise:

Referee #1: lipid metabolism/transport, biochemistry

Referee #2: lipid metabolism/transport, structural biology

Reviewers' Comments:

Reviewer #1:

Remarks to the Author:

Pascoa et al report cryoEM structures of human Ceramide synthase 6 (CerS6) in the absence and presence of the general CerS inhibitor fumonisin B1 (FB1). In the apo structure, there was a lipid like density occupying a narrow internal cavity. Based on mass spectrometry the authors propose this represents an intermediate stage of catalysis with palmitoyl covalently bonded to a nucleophilic Histidine. The FB1 structure sees this density now covalently bonded to FB1, which is proposed to be a N-palmitoyl-FB1 moiety. Again mass spectrometry is used to support this finding. Based on the structures the authors propose a ping-pong catalytic mechanism for CerS enzymes where the acyl-CoA substrate first forms a covalent bond with the catalytic Histidine residue, and then the second incoming sphinganine substrate acts as nucleophile to form dihydroceramide.

The basic conclusions appear to be supported by the data and the structures represent a major breakthrough with regard to CerS structure-function, especially given CerS6 is a therapeutic target for metabolic disease. However, some issues were noted that should be addressed. In particular, clarification of the mass spec analysis with regards to palmitoyl-His species, double checking the modeling of key atomic interactions, and instances where the mutational analyses are not consistent with the proposed roles of residues in catalysis. In addition, integration of recent published studies on the structure and mutational analysis of yeast CerS should be considered and incorporated into the authors conclusions/interpretations.

Major points

1. Clarification is needed for how the theoretical mass shift for the covalent bond between His211 and palmitate was calculated. Why did the observed mass shift change between WT (238.45 or 238.53 Da (extended data 1) and the non-glycosylated N18Q mutant (239.8 Da)? Some clarification is needed as my crude estimate gives a mass shift of 239.42, closer to the non-glycosylated protein). Given the importance of the identity of this covalently bound lipid, it is critical to explain these calculations in detail.
2. Were MS experiments carried out in the presence or absence of the Nb22 nanobody? If done in the presence of Nb22 they should be repeated to ensure the nanobody does not artificially induce trapping of the acyl-His intermediate.
3. Are the MS data deposited in any publicly available data bank (e.g. PRIDE)? This is important for data availability.
4. The structure of yeast Lac1p bound to an acyl-coA (ref 56, EMBO) is mentioned in the discussion section but there is no discussion as to why that structure bound intact acyl-coA and did not form an acyl-His intermediate.
5. The statement that the central cavity has a large cytoplasmic opening (line 135) contradicts statements (lines 185-186 in results and lines 425-426) that there is not enough room to accommodate both substrates. How did the authors determine that there was not enough room for both substrates to bind simultaneously? Some clarification is needed. Also, did the authors also observe the same lateral opening seen in the Lac1p acyl-coA structure?
6. Palmitate structure modeling – The fit to the density of the last three atoms of palmitate (that are modeled to covalently bond to His211) is poor compared to the remaining aliphatic carbon chain of palmitate. This is reflected in the higher temp factors (~47) compared to the other palmitate atoms (~25). Are the authors confident in their current modeling in this critical position? It is noteworthy that the carbonyl oxygen is pointing in a different direction in the Palm structure vs the FB1 structure. I suggest reviewing the model building as this may explain the lack of any major effect from mutating Asn315.
7. Catalytic mechanism: why was the role of a nucleophilic histidine surprising (line 250)? Was this not expected based on sequence homology and prior experimental and structural studies? Are their conserved serine or cysteine residues that were previously implicated in catalysis?
8. The interpretation of mutational analyses with regard to specific roles of residues in the catalytic mechanism is lacking. As written, it is unclear how the authors propose the catalytic mechanism to occur beyond a basic ping-pong mechanism (Fig. 6), as there is no detailed chemistry involved in the proposed catalytic mechanism schematic, even though much of the mutational aspects of the manuscript focuses on this. For example, the N315D mutant appears to increase activity, which is contradictory for Asn315 to act as a hydrogen bond donor to stabilize the tetrahedral oxyanion state, and the corresponding Asn to Ala mutation in Lac1p also has the lowest impact on activity compared to other residues adjacent to the double His pair (ref 56). As noted above, the modeling of the carbonyl group of palmitate fits poorly to the density, which should be considered/noted. Mutation of R275 to lysine nearly abolishes activity (and the corresponding R318A mutation in Lac1p eliminates all activity) but it is simply stated that R275 is important to align neighboring residues for function. Ideally, given the proposed ping-pong catalytic mechanism, the authors should include a chemistry-based schematic of their proposed reaction mechanism that integrates their structure and mutational analyses, and also considers prior studies including the

acyl-CoA bound Lac1p structure/mutational analysis.

7. The methods for the thermal unfolding experiments are unclear. Did the authors monitor Tryptophan fluorescence or use a thiol reactive dye? More details are needed (e.g. excitation and emission wavelengths)

Minor issues:

Line 259- it is mentioned His212Ala mutation results in complete loss of activity. Was this or a similar mutation already demonstrated to result in loss of activity for a CerS enzyme? If so, it is worth citing/mentioning any relevant manuscript(s).

Lines 316-318: This sentence was found to be confusing. Try to reword to make clear that the new covalent bond being formed is between the palmitoyl group and FB1, and not FB1 and the enzyme.

It was unclear in the main text and main figure legend if the nanobody was used to determine the structure of CerS6 with FB1. The methods and extended Fig. 8 clarify this, but would be nice to mention the nanobody was also used for the FB1 structure in the main text for clarity (lines 304-305).

Mike Airola

Reviewer #2:

Remarks to the Author:

This is an interesting paper outlining not only the structure of a mammalian CerS but insights into some of the catalytic chemistry. The structure analyses is fairly standard with the use of GND and a camelid nanobody to assist in obtaining dimers with stable structures.

The newest, and perhaps most impactful data in this paper pertains to the identification of a potential catalytic mechanism. While the mammalian and yeast enzymes are overall identical, there are significant differences that give insight into potential differences in catalytic mechanisms. The yeast has a lateral opening in the Lac1p domain and there was a lipid-like density close to the opening suggesting it provided access for the sphingoid base in catalysis. In the present study, the structure of the mammalian CerS lacks this opening, but the authors did identify a density of in the structure consistent with the presence of N-palmitoyl and an inhibitor fumonisins B1. The authors suggest a ping pong reaction and this is supported by data showing palmitate bound to H211 which was eliminated when the enzyme was first incubated with the second substrate, sphinganine and led to the generation of C16:0 dihydroceramide. These, and additional, data provide a model for substrate entry and product release in a ping pong reaction. This will definitely be of interest to the readers of Nature Structural Biology.

There are no major concerns. It may have been helpful to provide an analyses of the enzyme kinetics to further support the ping pong mechanism but this is not a serious concern given the structural and mass spectroscopic data included in this manuscript.

Version 1:

Decision Letter:

Our ref: NSMB-A48502A

17th Apr 2024

Dear Dr. Sauer,

Thank you for submitting your revised manuscript "Structural basis of the mechanism and inhibition of a human ceramide synthase" (NSMB-A48502A). It has now been seen by the original referees and their comments are below. The reviewers find that the paper has improved in revision, and therefore we'll be happy in principle to publish it in Nature Structural & Molecular Biology, pending minor revisions to satisfy the referees' final requests and to comply with our editorial and formatting guidelines.

Sincerely,

Katarzyna Ciazynska, PhD
(she/her)

Associate Editor
Nature Structural & Molecular Biology
<https://orcid.org/0000-0002-9899-2428>

Reviewer #1 (Remarks to the Author):

The authors have addressed all my concerns and uncertainties. Notably, this includes conducting new high resolution MS experiments AND determining a new high res structure with a different nanobody that more clearly shows the connectivity between the His-palmitoyl linkage.

Reviewer #2 (Remarks to the Author):

This is a resubmission of an interesting paper outlining the structure of a mammalian CerS as well as provide insights into some of the catalytic chemistry. The structure analyses is fairly standard with the use of GND and a camelid nanobody to assist in obtaining dimers with stable structures. The newest, and perhaps most impactful data in this paper pertains to the identification of a potential catalytic mechanism. While the mammalian and yeast enzymes are overall identical, there are significant differences that give insight into potential differences in catalytic mechanisms. The yeast has a lateral opening in the Lac1p domain and there was a lipid-like density close to the opening suggesting it provided access for the sphingoid base in catalysis. In the present study, the structure of the mammalian CerS lacks this opening, but the authors did identify a density of in the structure consistent with the presence of N-palmitoyl and an inhibitor fumonisin B1. The authors suggest a ping pong reaction and this is supported by data showing palmitate bound to H211 which was eliminated when the enzyme was first incubated with the second substrate, sphinganine and led to the generation of C16:0 dihydroceramide. These, and additional data provide a model for-substrate entry and product release in a ping pong reaction. This will definitely be of interest to the readers of Nature Structural Biology. The only previous major concern was the suggestion that it may be helpful to provide an analyses of the enzyme kinetics to further support the ping pong mechanism. It was noted that this is not a serious concern given the structural and mass spectroscopic data included in this manuscript but the authors now note that while important and currently being planned, these studies go beyond the scope and key points of this manuscript and I agree. Overall this study represents a significant and original contribution that will have a significant impact on the field. The methods are appropriate and statistically significant.

Version 2:

Decision Letter:

1st Oct 2024

Dear Dr. Sauer,

We are now happy to accept your revised paper "Structural basis of the mechanism and inhibition of a human ceramide synthase" for publication as an Article in Nature Structural & Molecular Biology.

As soon as your article is published, you can generate your shareable link by entering the DOI of your article here: ><http://authors.springernature.com/share>. Corresponding authors will also receive an automated email with the shareable link

Your paper will be published online soon after we receive proof corrections and will appear in print in the next available issue. You can find out your date of online publication by contacting the production team shortly after sending your proof corrections.

An online order form for reprints of your paper is available at <[a href="https://www.nature.com/reprints/author-reprints.html"](https://www.nature.com/reprints/author-reprints.html)><https://www.nature.com/reprints/author-reprints.html>. Please let your coauthors and your institutions' public affairs office know that they are also welcome to order reprints by this method.

Please note that *Nature Structural & Molecular Biology* is a Transformative Journal (TJ). Authors may publish their research with us through the traditional subscription access route or make their paper immediately open access through payment of an article-processing charge (APC). Authors will not be required to make a final decision about access to their article until it has been accepted. <[a href="https://www.springernature.com/gp/open-research/transformative-journals"](https://www.springernature.com/gp/open-research/transformative-journals)> Find out more about Transformative Journals

Authors may need to take specific actions to achieve <[a href="https://www.springernature.com/gp/open-research/funding/policy-compliance-faqs"](https://www.springernature.com/gp/open-research/funding/policy-compliance-faqs)> compliance with funder and institutional open access mandates. If your research is supported by a funder that requires immediate open access (e.g. according to <[a href="https://www.springernature.com/gp/open-research/plan-s-compliance"](https://www.springernature.com/gp/open-research/plan-s-compliance)>Plan S principles) then you should select the gold OA route, and we will direct you to the compliant route where possible. For authors selecting the subscription publication route, the journal's standard licensing terms will need to be accepted, including <[a href="https://www.springernature.com/gp/open-research/policies/journal-policies"](https://www.springernature.com/gp/open-research/policies/journal-policies)>self-archiving policies. Those licensing terms will supersede any other terms that the author or any third party may assert apply to any version of the manuscript.

Sincerely,

Katarzyna Ciazynska, PhD
(she/her)
Senior Editor
Nature Structural & Molecular Biology
<https://orcid.org/0000-0002-9899-2428>

Reviewers' Comments:

Reviewer #1:

Remarks to the Author:

Pascoa et al report cryoEM structures of human Ceramide synthase 6 (CerS6) in the absence and presence of the general CerS inhibitor fumonisin B1 (FB1). In the apo structure, there was a lipid like density occupying a narrow internal cavity. Based on mass spectrometry the authors propose this represents an intermediate stage of catalysis with palmitoyl covalently bonded to a nucleophilic Histidine. The FB1 structure sees this density now covalently bonded to FB1, which is proposed to be a N-palmitoyl-FB1 moiety. Again mass spectrometry is used to support this finding. Based on the structures the authors propose a ping-pong catalytic mechanism for CerS enzymes where the acyl-CoA substrate first forms a covalent bond with the catalytic Histidine residue, and then the second incoming sphinganine substrate acts as nucleophile to form dihydroceramide.

The basic conclusions appear to be supported by the data and the structures represent a major breakthrough with regard to CerS structure-function, especially given CerS6 is a therapeutic target for metabolic disease. However, some issues were noted that should be addressed. In particular, clarification of the mass spec analysis with regards to palmitoyl-His species, double checking the modeling of key atomic interactions, and instances where the mutational analyses are not consistent with the proposed roles of residues in catalysis. In addition, integration of recent published studies on the structure and mutational analysis of yeast CerS should be considered and incorporated into the authors conclusions/interpretations.

Major points

1. Clarification is needed for how the theoretical mass shift for the covalent bond between His211 and palmitate was calculated. Why did the observed mass shift change between WT (238.45 or 238.53 Da (extended data 1) and the non-glycosylated N18Q mutant (239.8 Da)? Some clarification is needed as my crude estimate gives a mass shift of 239.42, closer to the non-glycosylated protein). Given the importance of the identity of this covalently bound lipid, it is critical to explain these calculations in detail.

We thank the reviewer for pointing out the small difference in the mass shifts for the observed palmitoylation of CerS6 across the various experiments.

The theoretical mass shift was calculated from the expected mass addition of a palmitoyl (C16:0) chain (average mass: 239.42 Da) with loss of a proton (average

mass -1.01 Da) required to protonate the leaving CoA-SH. This gives a theoretical value of +238.41 Da for the average mass shift for this covalent palmitoyl addition at His211. Taking together all our measurements of the covalent adduct present in the purified wild-type CerS6 samples (n=6), we experimentally determined the average mass shift for this modification to be $+238.52 \pm 0.07$ Da, which agrees with the proposed theoretical mass shift for this modification. We observed and noted this same effect in our study of ELOVL7 (Nie et al. 2021), which, like CerS6, also uses a ping-pong reaction mechanism with formation of a covalent acyl-enzyme intermediate at the first histidine of a histidine pair, as discussed in the manuscript. The mass shifts detected upon reaction of ELOVL7 with C18:0-CoA and C18:3(n3)-CoA also suggest covalent attachment of the acyl chain with loss of a proton.

Furthermore, we have verified the identification of the modification as a palmitoyl group through high-resolution mass spectrometry analysis. This was achieved by examining the enzymatic products formed after incubating the acyl-enzyme species with sphinganine, FTY720, and fumonisin B1. In every instance, the mass observed for the products agrees with the anticipated addition of a palmitoyl group to the primary amine of the substrate. This process benefited from the enhanced mass accuracy afforded by high-resolution mass spectrometry. We have revised the text to emphasize this point (lines 205-206).

Lastly, regarding the difference in the observed mass shift between the various replicates carried out using the wild-type and the N18Q mutant spectrum presented in ED Figure 1, it is important to consider the typical mass error range for intact protein mass spectrometry, which is often between 20 to 50 parts per million (p.p.m) (Chalk et al. 2017, Donnelly et al. 2019). Consequently, deviations within approximately 1 Da are entirely within the expected range for a protein the size of CerS6, which has a molecular weight of 44 kDa.

References:

Chalk R. Mass Spectrometric Analysis of Proteins. *Methods Mol Biol.* 2017;1586:373-95.

Donnelly DP, Rawlins CM, DeHart CJ, Fornelli L, Schachner LF, Lin Z, et al. Best practices and benchmarks for intact protein analysis for top-down mass spectrometry. *Nat Methods.* 2019;16(7):587-94.

2. Were MS experiments carried out in the presence or absence of the Nb22 nanobody? If done in the presence of Nb22 they should be repeated to ensure the nanobody does not artificially induce trapping of the acyl-His intermediate.

All MS experiments were carried out in the absence of nanobodies. We thank the reviewer for ensuring the rigor of our experiments and have revised the text to make this clear (lines 662-663).

3. Are the MS data deposited in any publicly available data bank (e.g. PRIDE)?

This is important for data availability.

The reviewer raises a key point about ensuring the availability of our MS data, and we are keen that our results enable further studies. However, we note that the PRIDE database is not well suited for our data, being aimed at proteomic results. Therefore, we have deposited all the raw mass spectrometry data into the Zenodo archive server, and included the DOI for this data in the revised manuscript (lines 797-798).

4. The structure of yeast Lac1p bound to an acyl-coA (ref 56, EMBO) is mentioned in the discussion section but there is no discussion as to why that structure bound intact acyl-coA and did not form an acyl-His intermediate.

The reviewer raises a key point of difference between our study of human CerS6 and the recent structure of yeast Lac1p. However, there are a number of key differences that preclude a direct comparison of the reports on these two homologs. Differences in the proteins' biochemical activity, host organism physiology, experimental conditions, and technical differences could all yield differences in the experimentally captured state.

We encountered challenges when comparing the structural results due to differences in map quality, particularly in the active site. Specifically, both of our maps (also see point #6) have unambiguous density for the acyl-His intermediate in the active site. In comparison, the map quality surrounding the acyl-CoA thioester in the Lac1p structure presents difficulties, appearing significantly less defined compared to the region around the end of the acyl chain. Furthermore, no biochemical studies were carried out on Lac1p to test for a covalent acyl-His intermediate of this protein.

Without equivalent studies being carried out, it is challenging to draw definitive conclusions about the reactivity of Lac1p with the acyl-CoA substrate. Teasing apart the similarities and differences between CerS6 and its yeast homologs will prove a fascinating avenue of further study, but will require a detailed examination of the physiology, methods, and structural results. While we have plans for such an analysis, it is best suited to a review article and beyond the scope of this manuscript.

5. The statement that the central cavity has a large cytoplasmic opening (line 135) contradicts statements (lines 185-186 in results and lines 425-426) that there is not enough room to accommodate both substrates. How did the authors determine that there was not enough room for both substrates to bind simultaneously? Some clarification is needed.

We thank the reviewer for raising a key point of our paper. The cytoplasmic opening which leads to the active site is indeed quite large. However, this narrows to 5Å in diameter just before the active site. This constriction sterically prevents both palmitoyl-CoA and sphingoid base from binding simultaneously. We have clarified this point in the revised text (lines 190-192).

Also, did the authors also observe the same lateral opening seen in the Lac1p acyl-coA structure?

Employing MOLE v2.5 with identical parameters confirmed the presence of this lateral opening in the structure of Lac1p, but we did not find a similar pathway in any of the CerS6 structures determined in this study. The mechanism of sphingoid base entry therefore remains unknown, and is the subject of planned future experiments.

6. Palmitate structure modeling – The fit to the density of the last three atoms of palmitate (that are modeled to covalently bond to His211) is poor compared to the remaining aliphatic carbon chain of palmitate. This is reflected in the higher temp factors (~47) compared to the other palmitate atoms (~25). Are the authors confident in their current modeling in this critical position? It is noteworthy that the carbonyl oxygen is pointing in a different direction in the Palm structure vs the FB1 structure. I suggest reviewing the model building as this may explain the lack of any major effect from mutating Asn315.

The reviewer raises an important point about the quality of the connectivity between the palmitoyl group and His211. This could be a consequence of inter-dimer movement, local motion in this region, and the inability of current cryo-EM data processing methods to address these subtle changes in the map(s).

To better resolve the palmitoyl-His211 connectivity, we have determined the structure of CerS6 in complex with a different nanobody (Nb02). The overall structure of CerS6 is effectively identical with both nanobodies (transmembrane region backbone RMSD = 0.39 Å). However, the connectivity of the palmitoyl-His211 is much better resolved in the CerS6-Nb02 complex, reflecting better nominal resolution, and local resolution in the active site of the protein.

Importantly, the cryo-EM map supports our initial modelling of the covalent linkage of the pamitoyl chain to His211. This additional map and model have been added to ED Fig 5 and ED Table 1, and the main text has been revised accordingly (lines 158-163).

7. Catalytic mechanism: why was the role of a nucleophilic histidine surprising (line 250)? Was this not expected based on sequence homology and prior experimental and structural studies?

We thank the reviewer for pointing out that the wording of our discovery of the reaction mechanism is not clear. Histidines are far less commonly employed as nucleophiles than cysteines or serines, though these have been found previously. We have reworded this section to clarify the point (line 256). Prior mutagenesis studies of ceramide synthase function (cited in the manuscript) had identified the essential role of the dihistidine motif. However, to our knowledge, ceramide synthases had never been proposed to use a nucleophilic histidine, and instead it was thought that these histidines could perhaps act in substrate binding and/or as general bases. Therefore, our findings on the role of these histidines in the catalytic mechanism of ceramide synthases, and form a covalent acyl intermediate, represent major developments in understanding the mechanism of these enzymes.

Are their conserved serine or cysteine residues that were previously implicated in catalysis?

We appreciate the reviewer's question regarding the potential role of conserved cysteine or serine residues in catalysis. The active site does not contain any cysteines, and it contains only one serine, Ser215. However, Ser215 is not conserved in any other human or yeast ceramide synthases. The nearest conserved serine, Ser222, is located towards the occluded end of the acyl-binding tunnel and is too distant (13 Å) from the active site to play a role in catalysis. Accordingly, this residue has been shown to be non-essential: mutation to an alanine does not affect the activity of human CerS5 and mouse CerS1 (Spassieva et al 2005 – ref. 17) or yeast Lag1 (Kageyama-Yahara and Riezman, 2006 – ref. 18).

8. The interpretation of mutational analyses with regard to specific roles of residues in the catalytic mechanism is lacking. As written, it is unclear how the authors propose the catalytic mechanism to occur beyond a basic ping-pong mechanism (Fig. 6), as there is no detailed chemistry involved in the proposed catalytic mechanism schematic, even though much of the mutational aspects of the manuscript focuses on this.

The reviewer raises a fascinating question regarding how ceramide synthases carry out our newly noted ping-pong reaction mechanism. This question prompts a deeper examination into the roles of specific residues within the active site. Despite our efforts to describe this mechanism through the design and analysis of the activity of various CerS6 mutants, significant questions remain.

For example, the N315D mutant appears to increase activity, which is contradictory for Asn315 to act as a hydrogen bond donor to stabilize the tetrahedral oxyanion state, and the corresponding Asn to Ala mutation in Lac1p also has the lowest impact on activity compared to other residues adjacent to the double His pair (ref 56). As noted above, the modeling of the carbonyl group of palmitate fits poorly to the density, which should be considered/noted.

Regarding the role of Asn315, our observation that the carbonyl oxygen of the palmitoyl group is within hydrogen bonding distance of its NH₂ (the modelling of the covalently attached palmitoyl chain has now been clarified in our answer to point #6), prompted us to consider its role in stabilizing the tetrahedral oxyanion transition state. This line of inquiry was further supported by the presence of either an Asn or a His at this position in other ceramide synthases. However, the finding that activity was preserved when Asn315 was mutated to His, Ala or Asp contradicts our initial hypothesis, leaving the specific role of Asn315 - and the mechanism by which the oxyanion transition state is stabilized - unresolved. We have revised the text to clarify this point (lines 285-286). Moreover, we thank the reviewer for referring to the mutant activity data provided in the Lac1p study (ref. 56), but would like to clarify that study's Asn296Ala mutant of Lac1p corresponds to Asn252 of CerS6, located at the cytoplasmic entrance to the central cavity, and not to Asn315.

Mutation of R275 to lysine nearly abolishes activity (and the corresponding R318A mutation in Lac1p eliminates all activity) but it is simply stated that R275 is important to align neighboring residues for function.

Our finding that the Arg275Lys mutation nearly abolishes all activity suggests that the mere presence of a positively charged side chain in this location in the active site is not sufficient for the catalytic function of CerS6. Our structures reveal that Arg275 forms hydrogen bonds with Asp242 and Asn315, leading us to propose that it may play a role in orienting neighbouring residues within the active site (lines 287-291 of the revised manuscript). While the precise manner in which this structural role of Arg275 impacts catalysis remains to be determined, it is plausible that it may relate to its interaction with Asp242 - essential for function in other ceramide synthases (refs. 17,18,56). However, exactly how these

residues participate in the catalytic mechanism also remains elusive, highlighting the need for further investigation.

Ideally, given the proposed ping-pong catalytic mechanism, the authors should include a chemistry-based schematic of their proposed reaction mechanism that integrates their structure and mutational analyses, and also considers prior studies including the acyl-CoA bound Lac1p structure/mutational analysis.

We thank the reviewer for this suggestion. However, as highlighted by the unresolved questions mentioned above, additional extensive and carefully designed experiments will be required to properly explore the CerS6 catalytic mechanism beyond our discovery here of CerS6's ping-pong mechanism. These are currently being planned, but are beyond the scope of this study. Therefore, until these experiments are completed, to include a detailed chemistry-based reaction mechanism with only the data in hand would be premature and potentially misleading. In light of these considerations, we have chosen to present the activity of our CerS6 mutants – while avoiding speculation on detailed mechanisms that do not accurately reflect our current level of understanding.

7. The methods for the thermal unfolding experiments are unclear. Did the authors monitor Tryptophan fluorescence or use a thiol reactive dye? More details are needed (e.g. excitation and emission wavelengths)

The thermal unfolding experiments were carried out using intrinsic tryptophan fluorescence. We thank the reviewer for pointing out this essential experimental parameter was unclear, and have revised the text to make this unambiguous (lines 567-568).

Minor issues:

Line 259- it is mentioned His212Ala mutation results in complete loss of activity. Was this or a similar mutation already demonstrated to result in loss of activity for a CerS enzyme? If so, it is worth citing/mentioning any relevant manuscript(s).

We thank the reviewer for pointing out the essentiality of noting prior work. Spassieva et al. (2006), Kageyama-Yahara et al. (2006), and Vanni et al. (2014) (refs. 17,18 and 47, respectively) carried out similar mutagenesis experiments, and we have tweaked the text (lines 266-267) to make clear how these works relate to our studies.

Lines 316-318: This sentence was found to be confusing. Try to reword to make

clear that the new covalent bond being formed is between the palmitoyl group and FB1, and not FB1 and the enzyme.

The reviewer is quite correct that this sentence is not very clear, and are very grateful for their note. We have modified this portion of the revised manuscript (lines 324-326) to make clear the reaction and connectivity.

It was unclear in the main text and main figure legend if the nanobody was used to determine the structure of CerS6 with FB1. The methods and extended Fig. 8 clarify this, but would be nice to mention the nanobody was also used for the FB1 structure in the main text for clarity (lines 304-305).

The same nanobody was used to determine both structures in our study. As Nb22 was only used as a fiducial marker for data processing, and the Nb22-CerS6 interface did not significantly change between structures, we elected to focus on the enzyme in our text and main figures. However, we are grateful for the reviewer pointing out that our enthusiasm to highlight the key scientific discoveries made our experimental method unclear. This has been clarified in the revised manuscript (line 310).

Mike Airola

Reviewer #2:

Remarks to the Author:

This is an interesting paper outlining not only the structure of a mammalian CerS but insights into some of the catalytic chemistry. The structure analyses is fairly standard with the use of GND and a camelid nanobody to assist in obtaining dimers with stable structures.

The newest, and perhaps most impactful data in this paper pertains to the identification of a potential catalytic mechanism. While the mammalian and yeast enzymes are overall identical, there are significant differences that give insight into potential differences in catalytic mechanisms. The yeast has a lateral opening in the Lac1p domain and there was a lipid-like density close to the opening suggesting it provided access for the sphingoid base in catalysis. In the present study, the structure of the mammalian CerS lacks this opening, but the authors did identify a density of in the structure consistent with the presence of N-palmitoyl and an inhibitor fumonisins B1. The authors suggest a ping pong reaction and this is supported by data showing palmitate bound to H211 which was eliminated when the enzyme was first incubated with the second substrate,

sphinganine and led to the generation of C16:0 dihydroceramide. These, and additional, data provide a model for-substrate entry and product release in a ping pong reaction. This will definitely be of interest to the readers of Nature Structural Biology.

There are no major concerns. It may have been helpful to provide an analyses of the enzyme kinetics to further support the ping pong mechanism but this is not a serious concern given the structural and mass spectroscopic data included in this manuscript.

We are grateful to the reviewer for pointing out that a detailed study of enzyme kinetics would greatly enrich the community's understanding of the CerS enzymatic mechanism. These studies are being planned now but go beyond the scope and key points of this manuscript.